# Defining the RNA interactome by total RNA-associated protein purification

Vadim Shchepachev[1], Stefan Bresson[1], Christos Spanos[1], Elisabeth Petfalski[1], Lutz Fischer[2], Juri Rappsilber[1,2,*] & David Tollervey[1,**]

## Abstract

The RNA binding proteome (RBPome) was previously investigated using UV crosslinking and purification of poly(A)-associated proteins. However, most cellular transcripts are not polyadenylated. We therefore developed total RNA-associated protein purification (TRAPP) based on 254 nm UV crosslinking and purification of all RNA–protein complexes using silica beads. In a variant approach (PAR-TRAPP), RNAs were labelled with 4-thiouracil prior to 350 nm crosslinking. PAR-TRAPP in yeast identified hundreds of RNA binding proteins, strongly enriched for canonical RBPs. In comparison, TRAPP identified many more proteins not expected to bind RNA, and this correlated strongly with protein abundance. Comparing TRAPP in yeast and *E. coli* showed apparent conservation of RNA binding by metabolic enzymes. Illustrating the value of total RBP purification, we discovered that the glycolytic enzyme enolase interacts with tRNAs. Exploiting PAR-TRAPP to determine the effects of brief exposure to weak acid stress revealed specific changes in late 60S ribosome biogenesis. Furthermore, we identified the precise sites of crosslinking for hundreds of RNA–peptide conjugates, using iTRAPP, providing insights into potential regulation. We conclude that TRAPP is a widely applicable tool for RBPome characterization.

**Keywords** mass spectrometry; phase separation; protein–RNA interaction; RNA binding sites; yeast

**Subject Categories** Methods & Resources; Post-translational Modifications, Proteolysis & Proteomics; RNA Biology

**Mol Syst Biol. (2019) 15: e8689**

## Introduction

Interactions between RNA and proteins play key roles in many aspects of cell metabolism. However, the identification of protein–RNA interaction sites has long been challenging, particularly in living cells. Individual protein–RNA interactions can be characterized, if known, by mutagenic and biochemical approaches, but this has always been labour-intensive. The difficulty is compounded by the fact that many interactions do not fall within characterized interaction domains, and even apparently well-characterized RNA binding domains can show multiple modes of RNA interaction (Clery *et al*, 2013), making detailed predictions less reliable. For large-scale characterization of protein–RNA interactions, a significant advance was the development of RNA immunoprecipitation (RIP) with or without formaldehyde crosslinking, allowing the identification of RNAs associated with target proteins, although not the site of association (Niranjanakumari *et al*, 2002; Gilbert *et al*, 2004; Hurt *et al*, 2004; Motamedi *et al*, 2004; Huang *et al*, 2005; Gilbert & Svejstrup, 2006). Subsequently, UV crosslinking approaches were developed that allow accurate identification of the binding sites for individual proteins on RNA molecules, transcriptome-wide (Maly *et al*, 1980; Wagenmakers *et al*, 1980; Mital *et al*, 1993; Urlaub *et al*, 2000; Rhode *et al*, 2003; Doneanu *et al*, 2004; Granneman *et al*, 2009, 2010; Bley *et al*, 2011; Van Nostrand *et al*, 2016). Two major approaches have been adopted for crosslinking. Short wavelength, 254 nm UVC irradiation can directly induce nucleotide–protein crosslinking and was used in initial crosslinking and immunoprecipitation (CLIP) analyses, as well as CRAC analyses in yeast. Subsequently, photoactivatable ribonucleoside-enhanced crosslinking and immunoprecipitation (PAR-CLIP) was developed, in which 4-thiouracil is fed to the cells and incorporated into nascent transcripts, allowing RNA–protein crosslinking to be induced by longer wavelength, ~350–365 nm UVA irradiation. Both approaches have been used extensively in many systems (reviewed in Darnell, 2010).

The reciprocal analyses of proteins that are bound to RNA were more difficult to develop, at least in part because proteomic approaches do not provide the amplification offered by PCR. However, UV crosslinking and RNA enrichment have been used to successfully identify many poly(A)$^+$ RNA binding proteins present in human cells and other systems (Baltz *et al*, 2012; Castello *et al*, 2012; Kwon *et al*, 2013). This technique was an important advance but, like RIP, identifies the species involved but not the site of interaction, and is limited to mature mRNAs. To identify the total RNA-bound proteome, the approach of 5-ethynyluridine (EU) labelling of

1   Wellcome Centre for Cell Biology, University of Edinburgh, Edinburgh, UK
2   Bioanalytics, Institute of Biotechnology, Technische Universität Berlin, Berlin, Germany
    *Corresponding author. Tel: +44 131 651 7056; E-mail: juri.rappsilber@ed.ac.uk
    **Corresponding author. Tel: +44 131 650 7092; E-mail: D.Tollervey@ed.ac.uk

RNAs followed by biotin ligation using the click reaction (RICK) was recently developed (Bao *et al*, 2018; Huang *et al*, 2018) as well as approaches based on phase separation, in which RNA–protein conjugates are recovered from an aqueous/phenol interface (Queiroz *et al*, 2019; Trendel *et al*, 2019). In addition, MS analyses have been developed to identify the precise amino acid at the site of RNA–protein crosslinking (Kramer *et al*, 2014).

The TRAPP techniques for identification of RNA–protein interaction are based on recovery of denatured RNA–protein complexes on silica beads, followed by mass spectrometry. They additionally permit identification of the peptide, and indeed the amino acid, that is crosslinked to RNA during UV irradiation in living cells. We applied TRAPP to identify RNA binding proteins from yeast and *Escherichia coli* following crosslinking in actively growing cells. These approaches should allow characterization of the protein–RNA interactome at steady state, as well as dynamic changes following stress exposure, in almost any system.

# Results

### Development of the TRAPP protocol

In all TRAPP techniques, we initially covalently linked all RNAs to associated proteins by UV crosslinking in actively growing cells. Detailed protocols for each of the TRAPP workflows are given in Materials and Methods.

In the TRAPP approach, ~750 ml cultures of actively growing yeast were irradiated at 254 nm (UVC). We initially irradiated with 1.4 J cm$^{-2}$, since similar doses have previously been used in many publications mapping protein-binding sites on RNA in yeast and *E. coli* (e.g. see Bohnsack *et al*, 2012; Sy *et al*, 2018). The workflow is outlined in Fig 1A. To quantify protein recovery by mass spectrometry in the presence and absence of UV crosslinking, the analyses incorporated stable isotope labelling with amino acids in cell culture (SILAC) (Ong *et al*, 2002) combined with the MaxQuant software package (Cox & Mann, 2008). For this, yeast strains that were auxotrophic for lysine and arginine (*lys9Δ0, arg4Δ0*) were grown in the presence of lysine and arginine or [$^{13}C_6$]-lysine plus [$^{13}C_6$]-arginine. Isotope-labelled and isotope-unlabelled cells were mixed after irradiation but prior to cell lysis. In all experiments, label swaps between crosslinked and non-crosslinked samples were included to confirm that the labelling did not affect the outcome of the analyses (Fig EV1A–C).

Briefly, irradiated cells were harvested by centrifugation, resuspended in buffer containing 2 M guanidine thiocyanate plus 50% phenol, and lysed by beating with zirconia beads. The lysate was cleared by centrifugation and adjusted to pH 4 by adding 3 M sodium acetate. The cleared lysate was incubated with silica beads in batch for 60 min. The beads were extensively washed in denaturing buffer: first in buffer containing 4 M guanidine thiocyanate, 1 M sodium acetate pH 4, then in low salt buffer with 80% ethanol. The ethanol wash is expected to reduce recovery of bound DNA (Avison, 2008). Nucleic acids and RNA-bound proteins were eluted from the column in Tris buffer. The eluate was treated with RNase A + T1 to degrade RNA, and proteins were resolved in a polyacrylamide gel in order to remove degraded RNA, followed by in-gel trypsin digestion and LC-MS/MS analysis.

The PAR-TRAPP approach is similar to TRAPP, except that cells were metabolically labelled by addition of 4-thiouracil (4tU) to the culture (final concentration 0.5 mM) for 3 h prior to irradiation, in addition to SILAC labelling. 4tU is rapidly incorporated into RNA as 4-thiouridine, thus sensitizing RNA to UVA crosslinking. However, 4tU strongly absorbs UV irradiation and confers considerable UV resistance on the culture (Fig EV1D). Cells were therefore rapidly harvested by filtration and resuspended in medium lacking 4tU immediately prior to irradiation at ~350 nm (UVA). Previous analyses using labelling with 4tU and UVA irradiation in yeast and other systems have generally involved significant crosslinking times (typically 30 min in a Stratalinker with 360 nm UV at 4 mJ s$^{-1}$ cm$^{-2}$), raising concerns about changes in RNA–protein interactions during this extended period of irradiation. We therefore constructed a crosslinking device (Fig EV1E) that delivers a substantially increased UV dose, allowing irradiation times of only 38 s to be used to deliver the equivalent dose of 7.3 J cm$^{-2}$. Subsequent treatment was as described above for TRAPP.

In principle, silica binding can enrich both RNA- and DNA-bound proteins, although UV crosslinking to dsDNA is expected to be inefficient and bound DNA should be reduced during ethanol washing of the silica beads (Angelov *et al*, 1988; Avison, 2008). To compare recovery of proteins bound to RNA versus DNA, samples recovered following initial silica binding and elution were treated with either DNase I, RNase A + T1 or cyanase (to degrade both RNA and DNA) and then rebound to silica, as outlined in Fig EV2A. Following washing and elution from the silica column, proteins were separated by SDS–polyacrylamide gel electrophoresis and visualized by silver staining (Fig EV2B and C). Nucleic acids were separated by agarose gel electrophoresis and visualized by SYBR safe staining to confirm degradation. Degradation of DNA had little effect on protein recovery, whereas this was substantially reduced by RNase treatment (Fig EV2D). The predominant proteins correspond to the added RNases. Cyanase-treated samples showed a low level of protein recovery (Fig EV2C), presumably due to residual nucleic acids surviving the treatment (Fig EV2D). We conclude that the TRAPP protocol predominately recovers RNA-bound proteins.

In the absence of UV irradiation (-UVC), a low level of protein recovery was also visible following RNA binding to silica (Fig EV2C). These proteins apparently bind RNA in the absence of crosslinking, even following denaturation, likely due to the mixed mode of RNA binding, involving both hydrophobic interactions with nucleotide bases and charge interactions with the phosphate backbone. These will be underestimated in SILAC quantitation of TRAPP analyses, potentially generating a small number of false-negative results. They can, however, be identified in iTRAPP (see below). We noted that a small number of proteins showed reduced recovery following UV exposure. We attribute this to crosslinking with other macromolecules, e.g. lipids, that are not retained by silica binding.

MaxQuant quantitation initially failed to return a value for many peptides in TRAPP MS/MS data, predominantly due to the absence of a detectable peptide in the −UV samples for the SILAC pairs (Fig EV3A–F). These "missing" peptides were strongly enriched for known RNA binding proteins, presumably because low abundance proteins that are efficiently purified are detected following crosslinking but not in the negative control. We

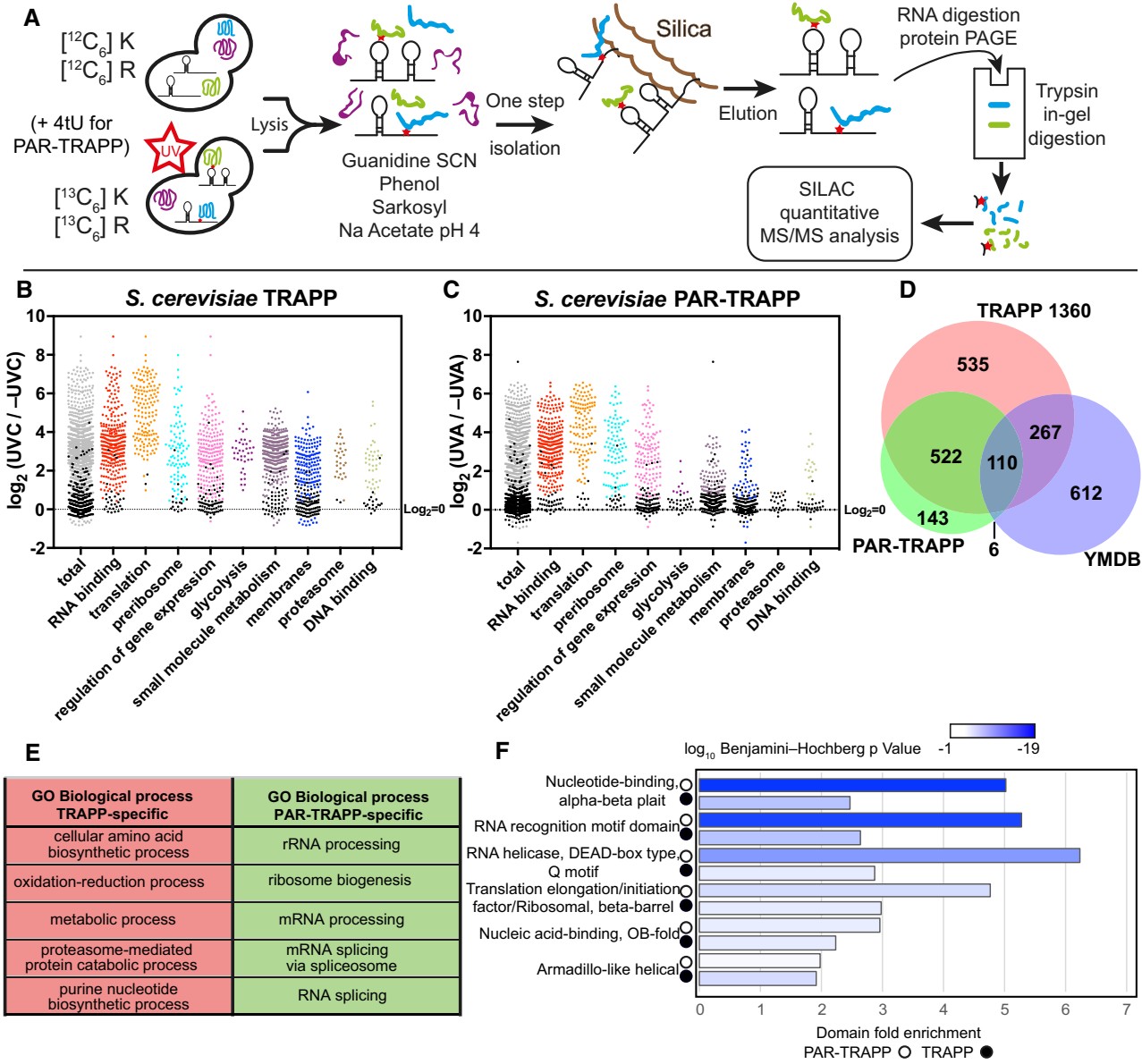

**Figure 1. TRAPP and PAR-TRAPP reveal the yeast RBPome.**

A  TRAPP and PAR-TRAPP workflows used to identify RNA-interacting proteins with SILAC MS-MS. See the main text for details.

B  Scatter plot of Log₂ SILAC ratios +UVC/−UVC (1,360 mJ cm⁻²) for *Saccharomyces cerevisiae* proteins, quantified with TRAPP. Proteins were subdivided based on the indicated GO term categories. Proteins belonging to GO terms "membrane" and "DNA binding" do not contain proteins mapping to GO terms "RNA metabolic process", "RNA binding", "ribonucleoprotein complex". Black dots represent proteins that failed to pass statistical significance cut-off (*P*-value adjusted < 0.05).

C  Scatter plot of Log₂ SILAC ratios +UVA/−UVA for *S. cerevisiae* proteins, quantified with PAR-TRAPP. Proteins were subdivided based on the indicated GO term categories. Proteins belonging to GO terms "small molecule metabolism", "membrane" and "DNA binding" do not contain proteins mapping to GO terms "RNA metabolic process", "RNA binding", "ribonucleoprotein complex". Black dots represent proteins that failed to pass statistical significance cut-off (*P*-value adjusted < 0.05). See Methods and Protocols for calculation of significance.

D  Venn diagram showing the overlap between proteins identified in TRAPP and PAR-TRAPP and proteins of intermediary metabolism annotated in the yeast metabolome database (YMDB).

E  5 most enriched GO terms amongst proteins identified only in TRAPP or exclusively in PAR-TRAPP.

F  6 most significantly enriched domains (lowest *P*-value) in PAR-TRAPP-identified proteins were selected if the same domain was enriched amongst TRAPP-identified proteins. Domain fold enrichment in the recovered proteins is plotted on the *x*-axis, while colour indicates log₁₀ Benjamini–Hochberg adjusted *P*-value.

addressed this problem by imputation of the missing values using the imputeLCMD R package (see Materials and Methods and Fig EV3G–O) (Lazar *et al*, 2016).

TRAPP with 1.4 J cm⁻² UVC in *Saccharomyces cerevisiae* identified 1,434 significantly enriched proteins, of which 1,360 were enriched more than twofold (Fig EV4A, Dataset EV1). Proteins

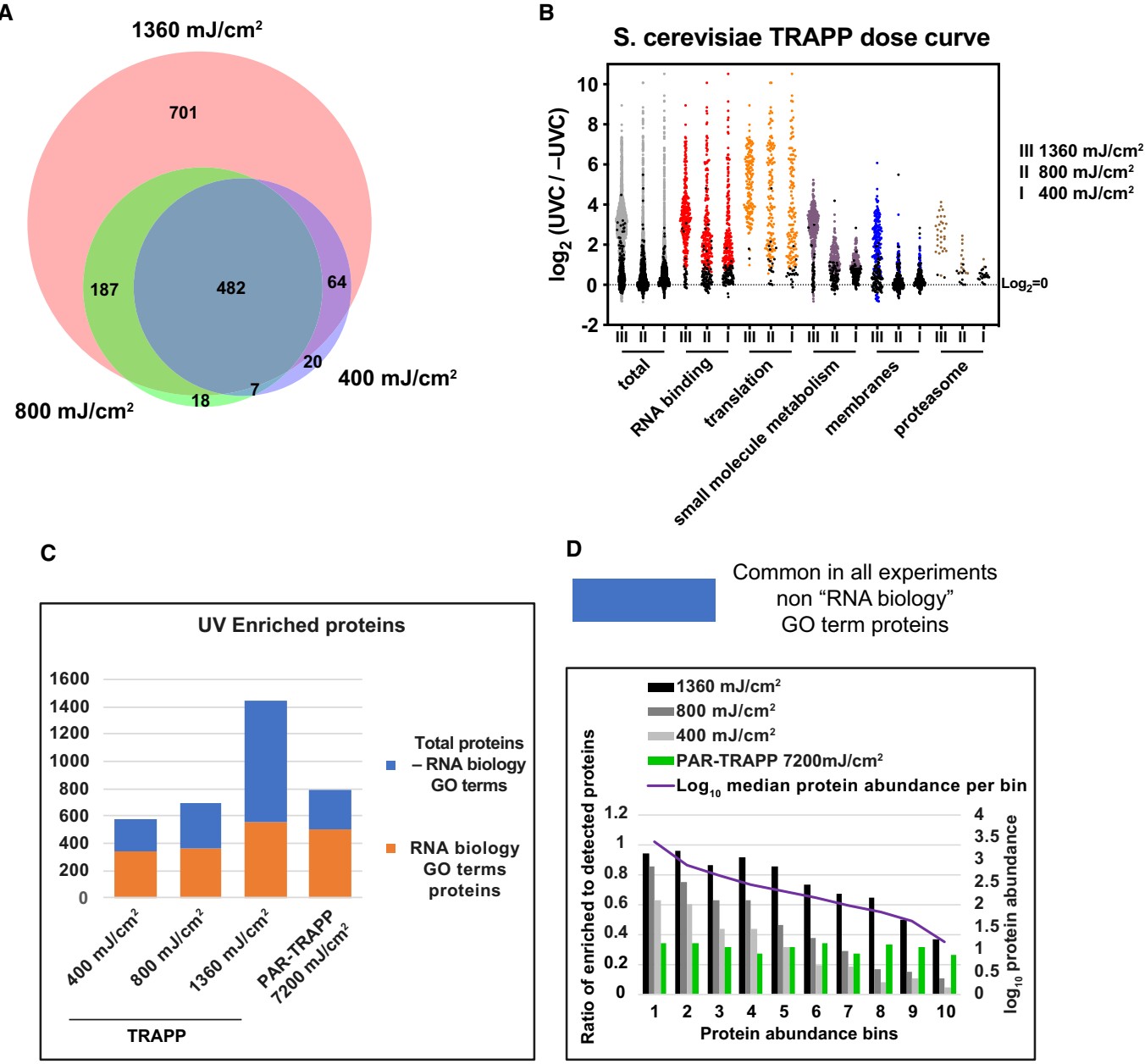

**Figure 2. The effect of UVC dose in *Saccharomyces cerevisiae* on the proteins identified in TRAPP.**

A  Venn diagram showing the overlap between proteins identified in TRAPP using the indicated UVC irradiation regime.

B  Scatter plot of Log₂ SILAC ratios +UVC/−UVC (for the indicated UV doses) for *S. cerevisiae* proteins, quantified with TRAPP. Proteins were subdivided based on the indicated GO term categories. Proteins, belonging to GO terms "membrane" and "small molecule metabolism" do not contain proteins mapping to GO terms "RNA metabolic process", "RNA binding", "ribonucleoprotein complex". Black dots represent proteins that failed to pass statistical significance cut-off (*P*-value adjusted < 0.05).

C  Proteins, identified in TRAPP and PAR-TRAPP were subdivided into 2 categories: "RNA biology" proteins (GO terms "RNA metabolic process", "RNA binding", "ribonucleoprotein complex") (orange bars); Proteins, not classified with either of the 3 GO terms above (blue bars). Numbers of proteins in each category are plotted per experiment.

D  Proteins quantified in both TRAPP and PAR-TRAPP were filtered to remove proteins annotated with GO terms "RNA metabolic process", "RNA binding", "ribonucleoprotein complex" (blue bars in Fig 2C). The remaining proteins were split into 10 bins by abundance (see Materials and Methods). For each bin, the ratio between enriched to detected proteins was calculated as well as median protein abundance as reported by PaxDb.

annotated with the GO term "translation" had the highest average fold enrichment, which is expected in the total RNA-interacting proteome (Fig 1B). It was previously reported that proteins involved

in intermediary metabolism can interact with RNA (e.g. see Beckmann *et al*, 2015; Queiroz *et al*, 2019; Trendel *et al*, 2019). Consistent with this, we observed significant enrichment for

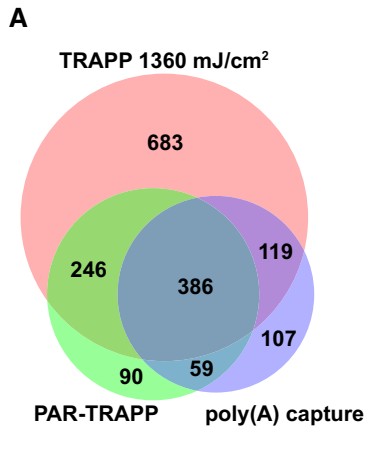

**A**

TRAPP 1360 mJ/cm²

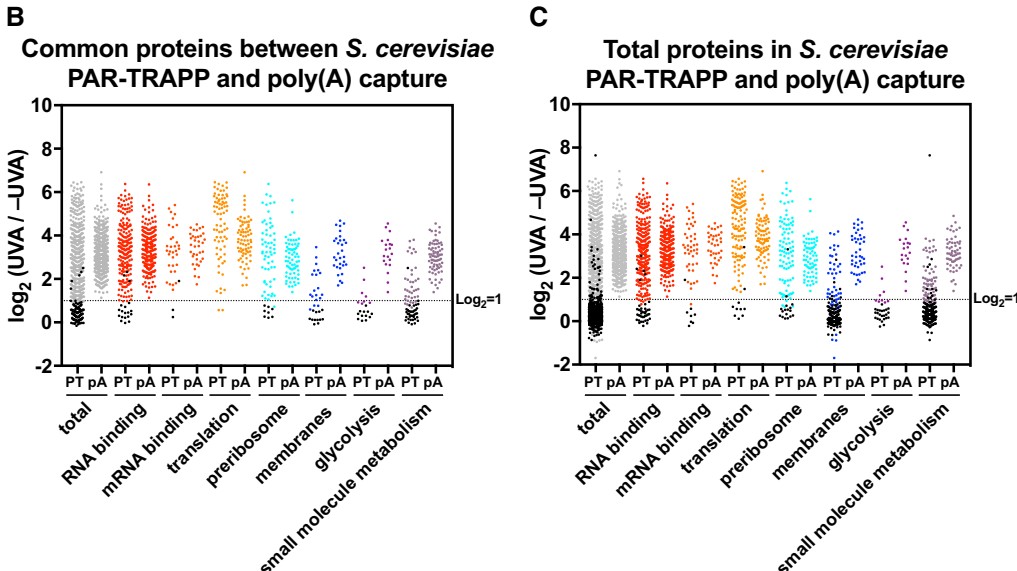

**Figure 3. The yeast RBPome identified by TRAPP compared to poly(A) RNA RBPome.**

A Venn diagram showing the overlap between proteins identified in PAR-TRAPP, poly(A) capture and TRAPP.

B Scatter plot of Log$_2$ PAR-TRAPP SILAC ratios +UVA/−UVA for *Saccharomyces cerevisiae* proteins and Log$_2$ +UVA/−UVA fold enrichment for poly(A) capture technique. Only proteins identified in both methods as RBPs are shown. Proteins were subdivided based on the indicated GO term categories. Proteins belonging to GO terms "membrane" and "small molecule metabolism" do not contain proteins mapping to GO terms "RNA metabolic process", "RNA binding", "ribonucleoprotein complex". Black dots represent proteins that failed to pass statistical significance cut-off (*P*-value adjusted < 0.05).

C Scatter plot of Log$_2$ SILAC ratios +UVA/−UVA for all *S. cerevisiae* proteins identified in PAR-TRAPP plotted together with Log$_2$ +UVA/−UVA fold enrichment for all proteins, reported as RBPs in poly(A) capture technique. Labelling is as in panel (B).

proteins annotated with GO term "small molecule metabolic process" (Fig 1B). Furthermore, 377 proteins enriched in TRAPP were annotated in the Yeast Metabolome Database (YMDB) (Ramirez-Gaona *et al*, 2017) to be enzymes or transporters associated with pathways of intermediary metabolism (Fig 1D). In particular, the majority of enzymes involved in glycolysis and/or gluconeogenesis were identified in TRAPP (Fig EV5) with average enrichment of more than fourfold. In addition, most of the structural components of the proteasome and many membrane-associated proteins were identified as RNA-binders by TRAPP.

Applying PAR-TRAPP identified twofold fewer significantly enriched proteins than found with TRAPP (Fig EV4B, Dataset EV2). However, PAR-TRAPP recovered a notably higher

proportion of characterized RNA binding proteins relative to proteins with less obvious connection to RNA. Only 116 (15% of total) proteins implicated in intermediary metabolism by the YMDB were identified amongst the PAR-TRAPP hits, relative to 377 (26% of total) in TRAPP (Fig 1D Dataset EV2). Furthermore, proteins annotated with the GO terms "glycolysis", "membrane part" and "proteasome" were substantially reduced relative to TRAPP (Fig 1C).

We performed GO term analyses on proteins that were exclusively found enriched in TRAPP or PAR-TRAPP (Fig 1D and E). The most enriched GO terms for TRAPP-specific proteins were related to metabolic processes and the proteasome, whereas PAR-TRAPP-specific proteins featured rRNA processing, mRNA

processing and RNA splicing. Furthermore, while both TRAPP and PAR-TRAPP demonstrated over-representation for known RNA-interacting domains within the enriched proteins (Dataset EV3), in PAR-TRAPP this trend was more pronounced both in

terms of higher fold enrichment and enrichment *P*-values (Fig 1F).

In these initial analyses, PAR-TRAPP clearly outperformed TRAPP. However, it seemed possible that the dose of 1.4 J cm$^{-2}$

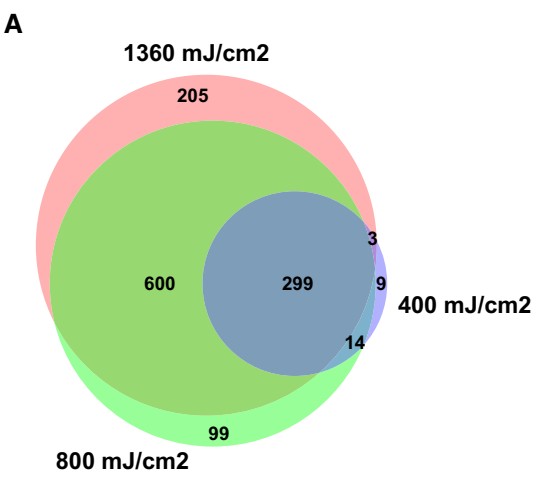

**A**

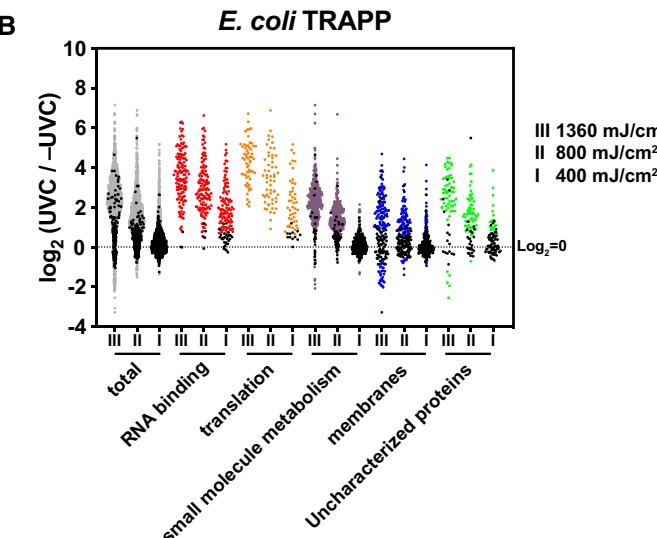

**B**

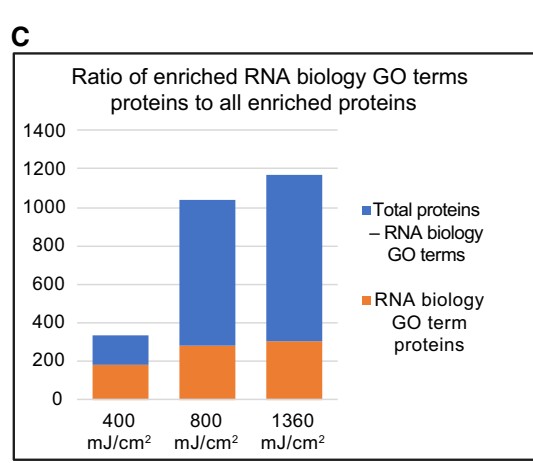

**C**

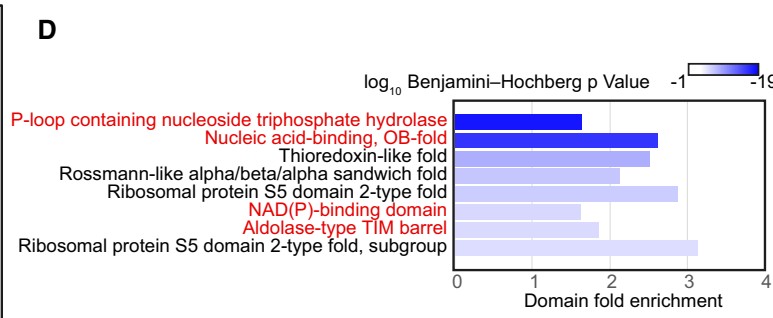

**D**

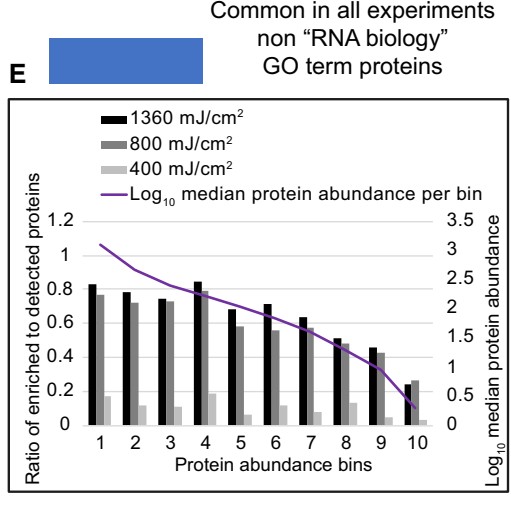

**E**

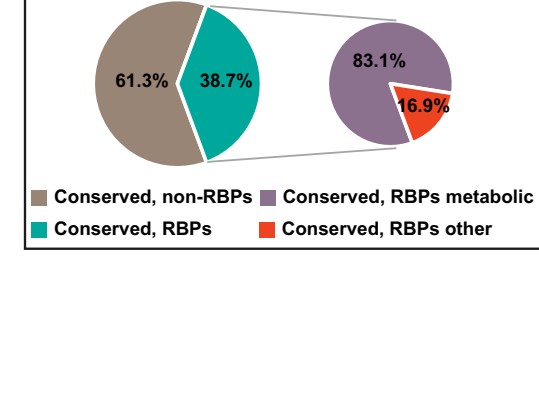

**F**

**Figure 4.**

**Figure 4. TRAPP reveals RNA binding proteins conserved from *Escherichia coli* to *Saccharomyces cerevisiae*.**

A Venn diagram showing the overlap between proteins identified in TRAPP using the indicated UVC irradiation regime.

B Scatter plot of $\log_2$ SILAC ratios +UVC/−UVC for *E. coli* proteins, quantified with TRAPP. Proteins were subdivided based on the indicated GO term categories. Proteins belonging to GO terms "membrane" and "small molecule metabolism" do not contain proteins mapping to GO terms "RNA metabolic process", "RNA binding", "ribonucleoprotein complex". Black dots represent proteins that failed to pass statistical significance cut-off (*P*-value adjusted < 0.05).

C Proteins, identified in *E. coli* TRAPP, were subdivided into two categories: "RNA biology" proteins (GO terms "RNA metabolic process", "RNA binding", "ribonucleoprotein complex") (orange bars); Proteins, not classified with either of the 3 GO terms above (blue bars). Numbers of proteins in each category are plotted per experiment.

D Most significantly enriched protein domains (lowest *P*-value) in *E. coli* TRAPP-identified proteins at $1,360 \text{ mJ cm}^{-2}$. Fold enrichment of the indicated domain amongst the recovered proteins is plotted on the *x*-axis, while colour indicates $\log_{10}$ Benjamini–Hochberg adjusted *P*-value. Domains found enriched in the yeast TRAPP data (at $1,360 \text{ mJ cm}^{-2}$) are labelled with red colour.

E Proteins quantified in all of the *E. coli* TRAPP experiments were filtered to remove proteins annotated with GO terms "RNA metabolic process", "RNA binding", "ribonucleoprotein complex" (blue bars in Fig 4C). The remaining proteins were split into 10 bins by abundance (see Materials and Methods). For each bin, the ratio between enriched to detected proteins was calculated as well as median protein abundance as reported by PaxDb.

F Pie chart of Inparanoid 8.0 database orthologous clusters between *S. cerevisiae* and *E. coli*. For a cluster to be labelled as conserved RNA interacting ("conserved, RBPs"), it was required to contain at least one bacterial and one yeast protein enriched in TRAPP (at $1,360 \text{ mJ cm}^{-2}$). "Conserved, RBPs metabolic" are clusters where at least one protein in yeast or bacteria is identified in the YMDB or in the ECMDB databases, respectively (see Materials and Methods).

UVC, previously optimized for recovery of RNAs bound to specific proteins, might not be optimal for recovery of the RNA-bound proteome. We therefore performed TRAPP using lower doses of 800 and $400 \text{ mJ cm}^{-2}$ UVC (Fig EV4C and D). As expected, reduced UVC exposure was associated with decreased numbers of statistically significant UV enriched proteins, with 482 core proteins enriched in TRAPP under all UVC exposure regimes (Figs 2A and EV4A, C and D). Reduced recovery was seen both for proteins annotated with selected "RNA biology" GO terms as well as amongst other proteins, such as metabolic enzymes and proteasome cofactors (Fig 2B and C). However, the latter proteins demonstrated higher attrition rate with reduction of UVC, presumably due to lower UV fold enrichment at the highest dose for these proteins in comparison with RNA biology" GO terms TRAPP hits. At the same time, amongst annotated, RNA-related proteins, there was relative enrichment for abundant, translation-related proteins (translation factors, ribosomal proteins), which were readily detected at all 3 UVC doses tested.

UVA crosslinking in PAR-TRAPP recovered similar number of annotated, RNA-related proteins as TRAPP with the highest dose of UVC, while recovery of other proteins was similar to TRAPP with the lowest exposure to UVC (Fig 2C). We assessed the correlation between cellular abundance and the likelihood of being reported as RNA-interacting protein by TRAPP and PAR-TRAPP. As TRAPP shows high enrichment for proteins involved in translation, "RNA biology" GO terms proteins were excluded from this analysis. Strikingly, this analysis revealed a clear trend for abundant proteins to be scored as enriched in TRAPP with 60–90% of proteins within the two most abundant bins were scored as enriched in +UVC, whereas no such correlation was observed in PAR-TRAPP data (Fig 2D). Only confidently identified proteins were included in each analysis, so this is unlikely to reflect a detection bias. Since the trend is not observed in PAR-TRAPP, it also seems unlikely to reflect a systematic bias towards higher confidence (lower *P*-values) for abundant proteins.

From these analyses, the optimal UVC exposure was unclear. Lower doses are potentially less noisy, but at the cost of recovering fewer proteins annotated functions in RNA biology, underlining the need for titration of the UV dose. As discussed below (see Discussion), TRAPP is predicted to better recover transient interactions than PAR-TRAPP due to differences in the UV–nucleotide

interactions involved. This suggests that high UVC exposure coupled with the sensitivity of TRAPP recovers transient RNA–protein contacts.

We further compared the RNA interactors identified in PAR-TRAPP with the published results of poly(A) capture (Beckmann *et al*, 2015), considering only proteins identified with $7.2 \text{ J cm}^{-2}$ UVA plus 4tU, as in PAR-TRAPP. There was a substantial overlap between proteins reported by PAR-TRAPP and poly(A) capture techniques, and this was increased by including the results of high UVC dose TRAPP (Fig 3A). First, we compared protein fold enrichment for selected GO terms in PAR-TRAPP versus poly(A) capture (Fig 3B and C). To avoid the bias of each RBPome isolation technique, this was performed only for the common proteins between the two methods (Fig 3B) as well as separately for all reported proteins (Fig 3C). A number of proteins annotated with GO terms "RNA binding", "translation", "preribosome" and "mRNA binding" had higher enrichment scores in PAR-TRAPP compared to poly(A) capture (Fig 3B). Considering that the administered doses of UVA in the two experiments are nearly identical, the difference may reside in the RNP purification approaches. This was notable for "mRNA binding" as poly(A) capture was expected to perform particularly well for these proteins. In contrast, the GO terms "membrane", "glycolysis" and "small molecule metabolic processes" showed higher average enrichment in poly(A) capture than PAR-TRAPP. This would be consistent with the proposal that at least some proteins of intermediary metabolism functionally interact with mRNA species (Beckmann *et al*, 2015).

## TRAPP identification of RNA binding proteins in Gram-negative bacteria

Sequence independent binding to silica, combined with aggressive, highly denaturing lysis, makes TRAPP potentially applicable to any organism. To confirm this, we applied TRAPP to *E. coli*, which is not amenable to proteome capture through poly(A) RNA isolation. We used TRAPP, as we could not identify a 4tU concentration that allowed both growth and efficient RNA–protein crosslinking. Initial crosslinking with ~$1.4 \text{ J cm}^{-2}$ UVC in *E. coli* identified 1,106 significantly enriched proteins, of which 1,089 were enriched more than twofold (Dataset EV4). Enrichment by GO term categories was similar to yeast TRAPP data (Fig 4A), with proteins involved in

**A**

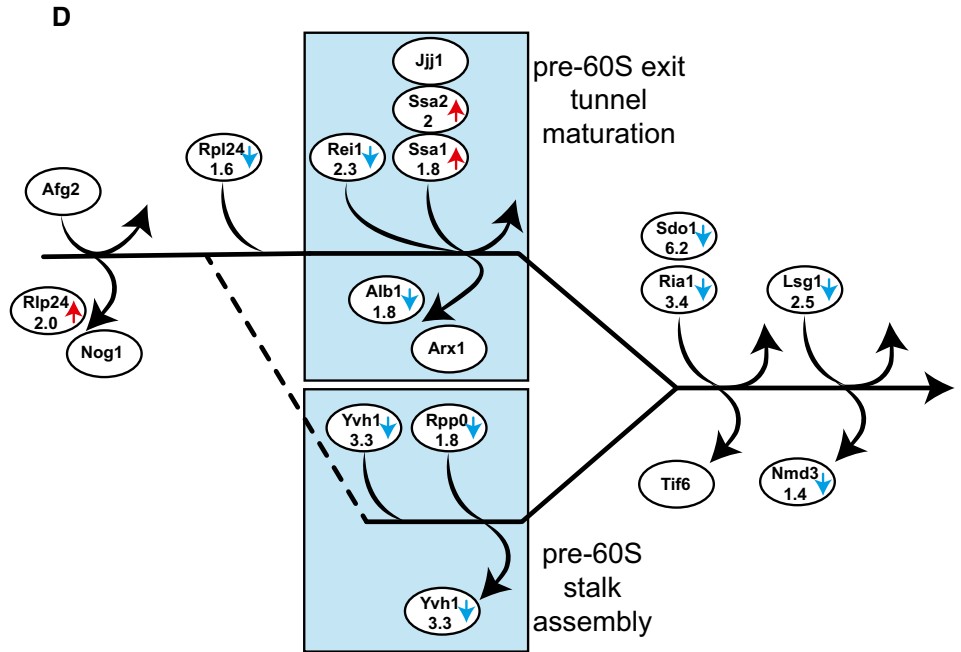

**B**

**C**

**D**

Figure 5.

**Figure 5.   TRAPP reveals the dynamics of RBPome upon stress.**

A   Volcano plot showing $\text{Log}_2$ protein abundance fold change in RBPome plotted against $-\text{Log}_{10}$ *P*-value. Black points represent proteins showing no statistically significant change upon sorbic acid exposure in PAR-TRAPP, while proteins changing significantly (*P*-value adjusted < 0.05) are labelled with blue. Only proteins observed as RNA interacting in PAR-TRAPP were included in the analysis.

B   Scatter plot of $\text{Log}_2$ SILAC ratios +Sorbic/−Sorbic for *Saccharomyces cerevisiae* proteins, quantified with PAR-TRAPP. Grey points represent proteins showing no statistically significant change upon sorbic acid exposure in PAR-TRAPP, while proteins changing significantly (*P*-value adjusted < 0.05) are labelled with other colours. Only proteins observed as RNA interacting in PAR-TRAPP were included in the analysis, except for proteins in the category "protein of intermediary metabolism extended", for which this criterion was dropped. Proteins annotated with GO term categories "RNA binding", "translation", "translation initiation", "P-body" and "small molecule metabolism" are displayed together with proteins annotated in literature-curated lists: "ribosome biogenesis 40S", "ribosome biogenesis 60S" (Woolford & Baserga, 2013). Proteins belonging to categories "protein of intermediary metabolism" and "protein of intermediary metabolism extended" are yeast enzymes and transporters of intermediary metabolism, obtained from YMDB and further filtered to remove aminoacyl-tRNA synthetases. "P-body core" category contains proteins identified as core components of P-bodies in yeast (Buchan *et al*, 2010). Numbers label the following protein on the chart: 1 − Rtc3; 2 − Tif3; 3 − Rpg1; 4 − Tif35; 5 − Gcd11; 6 − Rlp24; 7 − Ssd1; 8 − Rbp7; 9 − Nmd3.

C   Volcano plot showing $\text{Log}_2$ fold change in protein abundance upon sorbic acid exposure plotted against $-\text{Log}_{10}$ *P*-value. Black points represent proteins showing no statistically significant change in abundance upon sorbic acid exposure (*P*-value adjusted > 0.05).

D   The cytoplasmic phase of large subunit maturation in yeast (Lo *et al*, 2010). Proteins altered in abundance in PAR-TRAPP data upon sorbic acid exposure are indicated with arrows. Blue arrow denotes decrease, while red arrows indicate increase in PAR-TRAPP recovery upon stress. For proteins passing the statistical significance cut-off (*P*-value adjusted < 0.05), fold change is indicated.

translation showing the highest average enrichment compared to −UV. Other major RNA binding proteins showed enrichment similar to ribosomal proteins, including the RNA chaperone Hfq and the cold-shock proteins CspC, CspE, CspA and CspD.

Domain analysis showed enrichment for known RNA binding domains (Fig 4D). However, some domains not clearly related to RNA were also strongly enriched, notably NAD(P)-binding domains and the Aldolase-type TIM barrel domain, which were also identified in yeast TRAPP hits. Consistent with this, GO analyses identified many significantly enriched proteins categorized as "small molecule metabolism" (Fig 4B). Thioredoxin-like fold domains were enriched (Fig 4D), as previously found in human cells (Castello *et al*, 2016), suggesting that non-canonical protein–RNA interactions might be conserved.

This observation prompted us to analyse orthologous proteins between *S. cerevisiae* and *E. coli*. The Inparanoid 8 database (Sonnhammer & Östlund, 2015) reports 460 orthologous clusters between the two species, 39% of which had at least one bacterial and one yeast protein enriched in TRAPP (Fig 4F). Most of these clusters contained metabolic enzymes annotated in the corresponding metabolome databases for budding yeast and *E. coli*. We therefore extracted all conserved TRAPP proteins and performed a KEGG pathway enrichment analysis. This identified a number of enriched pathways (Appendix Fig S1A), including "glycolysis and gluconeogenesis" (Appendix Fig S1B) and "purine biosynthesis" (Appendix Fig S2), as well as "pentose phosphate".

The effects of altered UVC exposure were also assessed in *E. coli* (Figs 4A and EV4E–G). Overall fold enrichment was reduced with decrease in UV exposure (Fig 4B), both for proteins with annotated functions in RNA biology and proteins of intermediary metabolism. For annotated, RNA-linked proteins, 101 and 125 fewer were significantly enriched using 800 or 400 mJ cm$^{-2}$, respectively, relative to 1.4 J cm$^{-2}$ (Fig 4C). For proteins lacking RNA-related GO terms, we observed a correlation between abundance and TRAPP enrichment for the 2 highest UVC doses (Fig 4E). However, this correlation was not observed at 400 mJ cm$^{-2}$ irradiation, indicating that it is not intrinsic to the method. In contrast to yeast, the dataset of significantly enriched hits recovered with 400 mJ cm$^{-2}$ irradiation retained few metabolic enzymes (Fig 4B and E), emphasizing the importance of titration of UVC exposure.

## TRAPP reveals dynamic changes in RNA–protein interaction following stress

Cellular RNPs can be dramatically remodelled under altered metabolic conditions. For the initial analyses, we applied weak acid stress, which results in rapid cytoplasm acidification, as a drop in intracellular pH was reported to be a common feature of response to a variety of stresses (Weigert *et al*, 2009). Furthermore, reduced intracellular pH directly contributes to phase separation of poly(A) binding protein Pab1 into gel-like structures (Riback *et al*, 2017). SILAC-labelled yeast cells were treated with 6 mM sorbic acid for 16 min, to induce weak acid stress, followed by PAR-TRAPP.

Recovery of the majority of quantified proteins was unchanged upon this brief exposure to sorbic acid (Fig 5A and B). Statistically significant changes of > 2-fold were observed for 123 proteins; 99 (14%) showed reduced RNA association, while 23 (3%) were increased (Dataset EV5). The observed difference in PAR-TRAPP recovery of these proteins was not due to changes in protein cellular levels upon sorbic acid exposure (Fig 5C). Analysis of translation-related proteins indicated the specific inhibition of translation initiation, with six initiation factors showing > 2-fold reduced RNA binding. The most reduced were eIF4B (Tif3) (6.4-fold), eIF3A (Rpg1) (4.8-fold) and eIF3G (Tif35) (3.7-fold). Tif3 is of particular interest in the context of weak acid stress, since *tif3Δ* cells are hypersensitive to acetic and propionic acids (Kawahata *et al*, 2006; Mira *et al*, 2008) and a subset of yeast mRNAs are specifically sensitive to loss of Tif3 (Sen *et al*, 2016). We speculate that Tif3 may orchestrate responses to acid stress at the level of translation initiation.

The only translation initiation factor to show increased RNA-binding was the GTPase eIF2G (Gcd11), while the two other subunits of the eIF2 complex, eIF2A (Sui2) and eIF2B (Sui3), were unchanged. The role of eIF2 is to escort initiator methionine tRNA to 40S subunits during translation initiation. Similar enrichment upon sorbic acid exposure was observed for the translation elongation factor and GTPase, eEF1A (Tef1), which escorts tRNAs to elongating ribosomes. However, the results may reflect alternative functions for these GTPases during stress response, involving different interactions with RNA. For example, eEF1A was implicated in

tRNA export from the nucleus (Grosshans et al, 2000) and in the inhibition of Tor1 kinase activity, presumably acting together with uncharged tRNAs (Kamada, 2017).

A poorly characterized protein, Rtc3/Hgi1 showed sevenfold increased RNA binding upon sorbic acid treatment. Rtc3 was previously suggested to enhance translation of stress-response proteins (Gomar-Alba et al, 2012), and Rtc3 over-expression confers resistance to weak acids (Hasunuma et al, 2016).

Ribosome biogenesis also showed apparently specific changes following sorbic acid treatment (Fig 5B). Thirteen ribosome synthesis factors that are loaded onto cytoplasmic pre-60S particles following nuclear export showed reduced RNA interactions in sorbic acid-treated cells (Fig 5D). The nuclear–cytoplasmic shuttling, pre-60S factors Rlp24 and Nog1 are the first proteins to be released following nuclear export. RNA binding by Rlp24 was significantly increased in sorbic acid-treated cells (numbered 6 in Fig 5B); increased binding was also found for Nog1 but was not statistically significant. Rlp24 binds pre-60S particles in the nucleus, but following nuclear export, it is exchanged for the ribosomal protein Rpl24, which showed reduced RNA binding. This is mediated by the AAA-ATPase Afg2 (also known as Drg1) (Kappel et al, 2012). Displacement of Rlp24 is proposed to be a key initiation step for cytoplasmic pre-60S maturation (Pertschy et al, 2007). The only other cytoplasmic pre-60S factors, binding after Rlp24 release, with increased RNA binding were the homologous chaperones Ssa1 and Ssa2, potentially reflecting structural abnormalities, while all other quantified pre-60S maturation factors had reduced binding. We propose that weak acid stress induces a specific block in early, cytoplasmic pre-60S ribosomal subunit maturation (Fig 5D).

Following stress, mRNAs were reported to accumulate in storage and processing structures, termed P-bodies and stress granules (Decker & Parker, 2012), although effects of acid stress were not specifically assessed. It was, therefore, unexpected that reduced RNA interactions were seen for eleven proteins classed under the GO term "P-body". Exceptions included the global translation repressor Ssd1 (Kurischko et al, 2011; Hu et al, 2018), which showed > 2-fold increased RNA interaction, and Rpb7, which was identified in PAR-TRAPP exclusively under stress conditions. Rpb7 is a component of RNAPII, but dissociates following stress and participates in mRNA decay (Lotan et al, 2007; Harel-Sharvit et al, 2010), presumably explaining its detection only following sorbic acid treatment.

Amongst the 23 proteins showing > 2-fold increased RNA association following sorbic acid, eleven were enzymes participating in intermediary metabolism annotated in the YMDB (Ramirez-Gaona et al, 2017). In contrast, only two such enzymes were identified amongst the 100 proteins showing > 2-fold decreased RNA binding.

Our initial analysis only considered proteins that had been identified by PAR-TRAPP under non-stress conditions. Any proteins that interact with RNA only following exposure to stress would therefore be excluded, as found for Rpb7. To avoid this, we generated an extended list of all significant sorbic acid PAR-TRAPP hits. The extended list included 66 (4%) proteins with RNA interactions that increased > 2-fold following sorbic treatment, of which 36 were YMDB annotated metabolic enzymes, and 112 (7%) proteins with > 2-fold reduction, five of which were metabolic enzymes (Dataset EV5 extended list; Fig 5B). We conclude that multiple proteins of intermediary metabolism show substantially increased RNA association upon sorbic acid exposure. Notably, a set of metabolic enzymes were previously reported to accumulate in P-bodies and stress granules during stress response (Cary et al, 2015; Jain et al, 2016). Seven of these enzymes showed increased RNA interactions in PAR-TRAPP following sorbic acid treatment, while none showed reduced interactions.

### Identification of precise RNA binding sites in proteins by iTRAPP

Titanium dioxide ($TiO_2$) enrichment can be used to select RNA-crosslinked peptides species due to the presence of phosphate groups on RNA (Richter et al, 2009). In previous analyses (Kramer et al, 2014), and our initial analyses using 254 nm UV irradiation (preprint: Peil et al, 2018), the crosslinked nucleotide recovered was predominately uridine. For iTRAPP, cells were therefore incubated with 4tU and irradiated with 350 nm UV, as outlined in Fig 6A. Protein–nucleic acid conjugates were initially purified on silica as in TRAPP. Following elution, RNA was digested using nuclease P1, to leave nucleotide 5′ monophosphate groups, and proteins were fragmented with trypsin. Peptides crosslinked to phospho-nucleotides were recovered by enrichment on titanium dioxide ($TiO_2$) columns, and the site of crosslinking was identified by tandem MS (see Materials and Methods for further details).

Interpreting the spectra of an RNA-conjugated peptide represents a major challenge due to the fact that both RNA and peptide fragment generate rich ion spectra. Identifying the mass shift for peptides bound to fragmented RNA requires their annotation by specialist search engines. We compared the results of interpreting spectra using the published RNP[XL] pipeline (Veit et al, 2016) and the Xi search engine, which was designed for peptide–peptide crosslinks (Giese et al, 2015; preprint: Mendes et al, 2018) (https://github.com/Rappsilber-Laboratory/XiSearch). Better results were obtained with Xi (see Appendix Supplementary Methods), which was used for subsequent analyses. RNA modifications were identified with Xi search in a targeted-modification search mode (Dataset

**Figure 6. Identifying the RNA-crosslinked peptides with iTRAPP.**

A  iTRAPP workflow to directly observe crosslinked RNA–peptides species by mass spectrometry. See the main text for details.

B  Pie chart of RNA species observed crosslinked to peptides by the Xi search engine.

C  The analysis of amino acids, reported as crosslinked by the Xi search engine. Amino acids are represented by single letter IUPAC codes. Black bars—crosslink efficiency, defined as ratio between the frequency of the crosslinked amino acid and the frequency of the amino acid in all crosslinked peptides.

D  Venn diagram showing the overlap between proteins identified in PAR-TRAPP, RNP[xl] and Xi. Protein groups, reported by RNP[xl] and Xi, were expanded to single proteins so as to maximize the resulting overlap.

E  Domain structure of selected proteins, identified as crosslinked by the Xi search engine. Domains (coloured rectangles) and sites of phosphorylation (light green rhombi) from the UniProt database were plotted onto proteins represented by grey rectangles. Crosslink sites, identified by Xi, are indicated with red pentagons. See also Appendix Supplementary Methods.

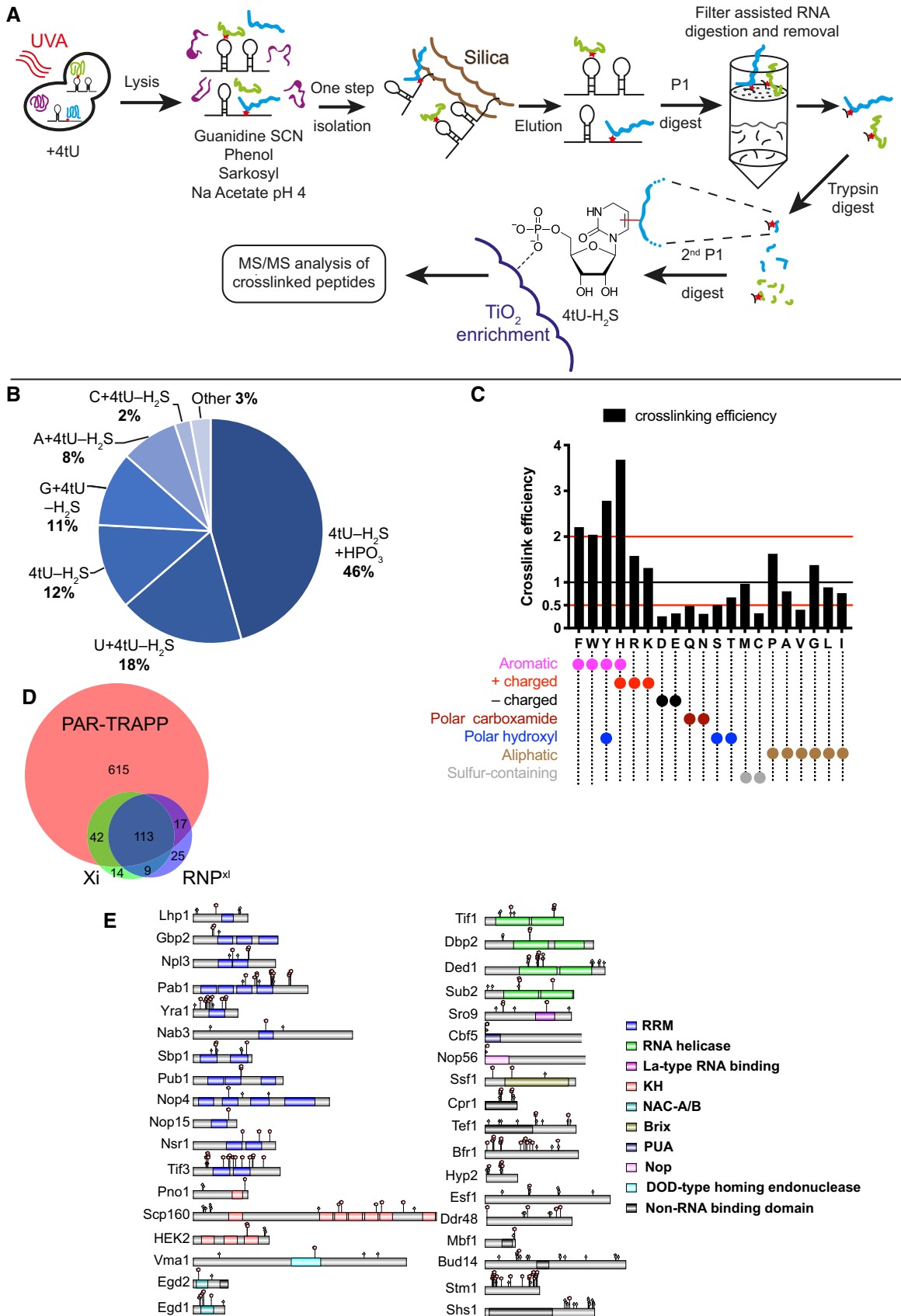

**Figure 6.**

EV6). Crosslinked nucleotides were treated as "cleavable molecules" during collision-induced dissociation in the gas phase. We generated search lists of 4-thiouridine-derived fragments allowed to persist on the peptide fragments upon fragmentation, without defining amino acid specificity (See Materials and Methods). The number of peptides to spectrum matches passing the 1% FDR cut-off was used to identify the optimal fragment list.

As previously observed (Kramer *et al*, 2011, 2014), the bulk of crosslinked nucleotides contained 4-thiouridine, lacking the mass of $SH_2$ (Fig 6B). Surprisingly, the most frequent RNA adduct was 4-thiouridine-containing 2 phosphate groups ($4tU-H_2S+HPO_3$), while nuclease P1, used for RNA degradation, generates 5′ monophosphate nucleotides. This may be a result of degradation of crosslinked dinucleotides during sample preparation or mass spectrometry.

It was formally possible that diphosphates-RNA-containing crosslinks represent preferential isolation of monophosphate-RNA crosslinked to phosphorylated peptides. Were the MS2 fragment coverage to be insufficient, these could potentially be miscalled as peptides crosslinked to a nucleotide 5′,3′ diphosphate. To test this, we utilized Lambda phosphatase (λ), which is active against a broad range of phosphorylated protein substrates but has limited activity towards nucleotides (Keppetipola & Shuman, 2007). Including treatment with λ phosphatase in the iTRAPP protocol decreased the number of conventional phospho-peptides reported by MaxQuant 10-fold compared to the untreated sample (Fig EV6A); however, the major crosslinked nucleotide remained the diphosphorylated species (Fig EV6B and Dataset EV7).

With the best performing search parameters and combining three datasets, keeping the highest scoring PSMs, we identified 524 unique RNA–peptide crosslinks with 418 different peptides belonging to 178 proteins (Dataset EV7). Notably, 50% of the observed crosslinks overlapped with protein domains previously implicated in RNA binding (Fig 6E).

The crosslinking efficiency for each amino acid was determined from the ratio between the abundance of the amino acid in the pool of crosslinked peptides and its frequency at the crosslink site (Fig 6C). This identified tryptophan (W), tyrosine (Y), phenylalanine (F), histidine (H), arginine (R), proline (P), lysine (K) and glycine (G) as preferentially crosslinked to RNA. Even for amino acids that were infrequently found at the crosslinking site, we could identify high-quality spectra with unambiguous mapping, demonstrating that these crosslinks are detectable when present (Appendix Figs S3–S5).

Thus, aromatic residues tend to have a clear preference for crosslinking, presumably due to stacking interactions with the bases. Positively charged amino acids, recognized by trypsin protease, also showed elevated crosslinking. The presence of a negatively charged RNA near trypsin cleavage site is expected to interfere with trypsin digest of the crosslinked protein. Indeed, 79 of crosslinked RNA–peptide crosslinks had 1 or 2 missed trypsin cleavages. This number rose to 97% for peptides in which Xi reported lysine or arginine as the crosslinked amino acid, indicating that trypsin cleavage is unlikely to occur at or near to the site of crosslinking.

It might be anticipated that charge repulsion would tend to block interaction between RNA and amino acids with closely spaced phosphorylation sites. Comparison of iTRAPP crosslinks with published sites of protein phosphorylation from all datasets revealed substantial enrichment for colocalization within 20 amino acids ($P = 2.3 \times 10^{-5}$). We speculate that phosphorylation of these sites will predominately be associated with the RNA unbound populations of these proteins, potentially regulating association.

Proteins identified by iTRAPP show significant, but not complete, overlap with PAR-TRAPP (Fig 6D). Out of 178 proteins with crosslinking sites mapped by Xi, 25 were not identified as significantly enriched in TRAPP. These included several characterized RNA binding proteins (Dataset EV8): e.g. rRNA methyltransferase Nop2, RNA surveillance factor Nab3 and poly(A) polymerase Pap1. These proteins likely retain RNA binding activity even under denaturing conditions leading to poor UV enrichment, but can be positively identified in iTRAPP.

### Eno1 binds RNA *in vivo*

Many metabolic enzymes showed greater recovery with TRAPP than PAR-TRAPP (see above). We therefore wanted to determine whether TRAPP-enriched factors are *bona fide* RNA binding proteins using a different technique. For this purpose, we selected the yeast glycolytic enzyme Eno1 (enolase), which was robustly identified by TRAPP, but showed enrichment of less than twofold in PAR-TRAPP. Enolase was also confidently identified as RNA binding by TRAPP in *E. coli,* as expected due to its presence in the RNA degradosome complex (Carpousis, 2007), and the homologue, Eno2, was previously reported to interact with mitochondrial tRNAs (Entelis *et al*, 2006).

To tag Eno1, the chromosomal *ENO1* gene was fused to a tripartite C-terminal tag, His6-TEV protease cleavage site-protein A (HTP), and the fusion protein was expressed from endogenous $P_{ENO1}$. The strain expressing only Eno1-HTP showed wild-type growth, indicating that the fusion protein is functional. This strain was then used in CRAC analyses, alongside the non-tagged negative control strain. In CRAC, proteins are subject to multistep affinity purification that includes two fully denaturing steps. Eno1-HTP was efficiently crosslinked to RNA in actively growing cells using 254 nm UVC. Figure 7A shows [5′ $^{32}$P] labelling of the RNA-associated with purified Eno1, relative to the background in a non-tagged strain. High-throughput sequencing of cDNAs generated from RNAs bound to Eno1 (Fig 7B and Dataset EV9) showed a wide distribution of target classes.

Amongst the top mRNAs bound by Eno1, the only overall enrichment was for highly expressed genes (Dataset EV9). Moreover, inspection of the hit distribution across all mRNAs identified with good sequence coverage did not indicate any specific binding site, which might have been expected for a regulatory interaction.

The distribution of Eno1 binding did appear to show greater specificity on tRNAs, which comprised ~25% of the recovered sequences (Fig 7B). The structure and modifications present in tRNAs reduce their recovery by standard RT–PCR, so this recovery is likely to underestimate the frequency of Eno1-tRNA interaction *in vivo*. Recovery was specific for cytoplasmic tRNAs relative to mitochondrial tRNAs (Fig 7D). Within tRNAs, Eno1 binding sites showed apparent specificity. For example, the crosslinking site in tRNA$^{Ala}_{UGC}$ was localized to a single nucleotide within the T-loop (Fig 7C) and similar patterns were seen on many tRNAs (Fig 7E and F).

These results indicate that Eno1 contacts specific structural elements within cytoplasmic tRNAs and highlight the potential for TRAPP to uncover novel RNA interactions.

                                    

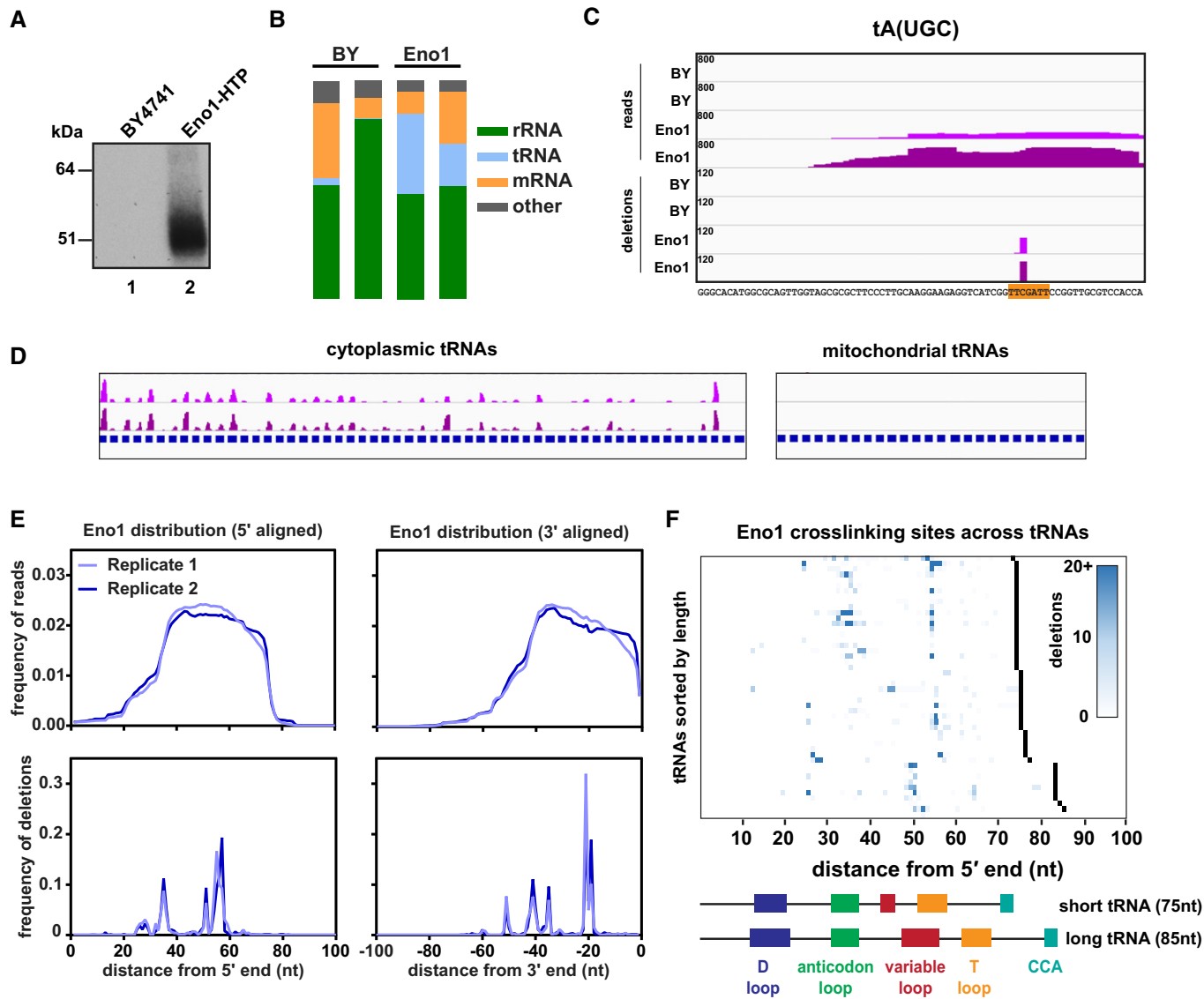

**Figure 7. RNA binding by Eno1.**

A Representative gel showing the recovery of radiolabelled RNA after CRAC purification. Lane 1: Untagged control strain (BY4741). Lane 2: Strain expressing Eno1-HTP from its endogenous locus.

B Bar charts showing the relative distribution of reads amongst different classes of RNA.

C The binding of Eno1 to the representative tRNA tA(UGC). The four upper tracks show the distribution of entire reads, while the four lower tracks show putative crosslinking sites (deletions). Tracks are scaled by reads (or deletions) per million, and this value is denoted in the upper left corner of each track. Two independent replicates are shown for the untagged BY control and Eno1-HTP. The tRNA sequence is shown below with the T-loop sequence highlighted in orange.

D Global view of Eno1 binding to cytoplasmic tRNAs (left) or mitochondrial tRNAs (right). For ease of viewing, tRNAs across the genome were concatenated into a single "chromosome", with each tRNA gene annotation shown in blue. Two independent replicates are shown.

E Metagene plots showing the distribution of reads or deletions summed across all tRNAs. tRNA genes were aligned from either the 5′ end (left) or the 3′ end (right).

F Heat map showing the distribution of putative crosslinking sites (deletions) across all Eno1-bound tRNAs. tRNA genes are sorted by increasing length, and the 3′ end for each gene is denoted in black. The domain structures of a typical short (tH(GUG)) and long (tS(AGA)) tRNA are included for comparison.

# Discussion

The identification of ever-increasing numbers of RNA species has underlined the importance of robust characterization of *bona fide* sites of protein–RNA interaction. Here, we report the development and application of protocols for the purification of the RNA-bound proteome, providing quantitative data on protein crosslinking efficiencies under normal or stress conditions, and identifying precise sites of protein–RNA contact.

**Differences between TRAPP and PAR-TRAPP results**

An unexpected finding was the differences in protein recovery in TRAPP using 254 nm UVC irradiation compared to PAR-TRAPP

using 4-thiouracil incorporation combined with long-wavelength UVA irradiation. In both cases, the most enriched classes of proteins recovered correspond to expected targets, including the translation machinery and many other annotated RNA-interacting proteins. However, in agreement with a previous report (He *et al*, 2016) there were clear differences in the recovery of proteins that lack known functions in RNA metabolism, which were much more abundant in TRAPP than with PAR-TRAPP.

The basis of the difference between crosslinking with 254 nm UV compared to 4-thiouracil with UVA may lie in the properties of the nucleotides. Native nucleotides show very rapid UV radiation decay with subpicosecond lifetimes (i.e. $< 10^{-12}$ s) (reviewed in Beckstead *et al*, 2016). This likely reflects non-radiative decay of the activated state, with the energy dissipating as heat, protecting the nucleotide from damage. This may be a feature that was selected early in the development of life, presumably in a very high UV environment (Beckstead *et al*, 2016). We speculate that during 254 nm irradiation, nucleotides can absorb many photons but will normally very rapidly release the energy as heat. However, if in close contact with an amino acid, crosslinking can occur during this short time window. This potentially allows most nucleotides to survive the entire UVC irradiation, while presenting multiple opportunities for crosslinking with transiently interacting proteins, although the frequency will be low for any individual nucleotide. In contrast, 4-thiouridine absorbs UVA and generates a reactive triplet with high quantum yield (0.9) (Zou *et al*, 2014). The 4-thiouridine can be crosslinked to an amino acid, if suitably positioned at that instant. Alternatively, a highly reactive singlet oxygen can be generated, which reacts with 4-thiouridine to generate uridine or uridine sulfonate (U-SO$_3$). In either case, no further crosslinking can occur. Our interpretation of the difference between the TRAPP and PAR-TRAPP data is, therefore, that 254 nm UV allows multiple chances of crosslinking during the period of irradiation, increasing the probability of capturing weak, transient interactions. In contrast, 4-thiouridine reacts with UV only once and is therefore biased towards the recovery of stable interactions.

This hypothesis leads us to propose that proteins specifically identified by TRAPP that lack known RNA-related functions and were not significantly enriched by PAR-TRAPP, will generally form weak, transient RNA interactions.

## Potential significance of "unexpected" RNA-interacting proteins

We noted that metabolic enzymes were major targets in TRAPP, including almost all glycolytic enzymes. There is every reason to think that these data capture genuine, *in vivo* RNA–protein contacts. In our data, many proteins were recovered reproducibly and significantly in TRAPP at all UVC doses tested and in PAR-TRAPP. Moreover, similar interactions have been independently identified in other analyses of RNA binding proteomes using UVC irradiation (Baltz *et al*, 2012; Castello *et al*, 2012; Kwon *et al*, 2013; Kramer *et al*, 2014). In the case of Eno1, we further confirmed its robust and apparently specific *in vivo* crosslinking.

Notably, there was strong conservation of RNA binding between homologous enzymes of intermediary metabolism yeast and *E. coli*, which is generally a sign of a conserved feature. We initially applied a UVC dose similar to that previously used for many studies

mapping protein-binding sites on RNA (1.4 J cm$^{-2}$). Applying lower UV doses decreased enrichment, both for proteins with annotated links to RNA biology and the less expected proteins of intermediary metabolism. For yeast, we identified no level of UVC irradiation that was clearly optimal. Lower doses recovered fewer proteins that are not clearly linked to RNA metabolism, but many known RNA-interacting proteins were lost, with an increased bias towards translation-related proteins. For *E. coli*, the lowest dose tested (400 mJ cm$^{-2}$) recovered relatively few proteins that were not previously linked to RNA metabolism. This may represent a good starting point for future analyses, given that we were unable to identify suitable conditions for PAR-TRAPP in *E. coli*.

We note that 400 mJ cm$^{-2}$ is lower than used in previously reported analyses of RNA–protein crosslinking in liquid culture. It is also notable that this dose is delivered to cells suspended in a volume of media surrounding a UVC source. It must be anticipated that for irradiation of a monolayer of cells attached to plates the effective UV dose delivered to cells will be substantially higher. Dose–response curves may therefore be necessary for future analyses of the RNA-bound proteome using UVC.

There has been substantial recent interest in the phenomenon of liquid–liquid phase separation or demixing (reviewed in Boeynaems *et al*, 2018). Different RNAs, proteins and RNP complexes can be greatly concentrated by these processes (e.g. see Clemson *et al*, 2009; Lin *et al*, 2015; Riback *et al*, 2017; Langdon *et al*, 2018; Maharana *et al*, 2018), presumably enhancing the efficiency of RNP assembly and processing (Lewis & Tollervey, 2000). Moreover, weak interactions between RNA and metabolic enzymes have been proposed to segregate functionally related enzymes into spatially confined "metabolons", potentially boosting the metabolic flux (Castello, 2015 #4733). However, uncontrolled self-assembly of abundant metabolic proteins into separated cellular domains seems highly undesirable. Since phase separation is driven by self-assembly, it can be opposed by the formation of large numbers of different, non-self-interactions with abundant structured RNA species (see Maharana *et al*, 2018). We therefore postulate that many of the novel interactions do not have specific functions in RNA biology. This does not, of course, preclude the possibility that individual, novel RNA–protein interactions do play specific and important roles.

To test this model, we analysed the major glycolytic enzyme Eno1, which gave substantially stronger crosslinking in TRAPP than in PAR-TRAPP. This revealed preferential interaction of Eno1 with the 3′ regions of cytoplasmic tRNAs, which are suitable RNAs to interact with to achieve an anti-phase separation effect. Notably, several other glycolytic enzymes were reported to form phase-separated, cytoplasmic "G bodies" following hypoxic stress (Miura *et al*, 2013; Jin *et al*, 2017). It seems possible that formation of these bodies might be regulated by protein–RNA interactions.

## PAR-TRAPP identifies stress-induced changes in RNA–protein interactions

We took advantage of the ease and reproducibility of TRAPP approaches to analyse the immediate effects of weak acid stress on the RNA-bound proteome. Analyses were performed comparing non-stressed cells, to cells treated with sorbic acid for 16 min, to reduce the intercellular pH. Yeast generally grows in acid

conditions, and reduced intracellular pH is a common feature in response to multiple stresses, including glucose deprivation, due to inactivation of the membrane proton pumps that maintain the pH gradient. Moreover, reduced pH was reported to solidify the cytoplasm, due to charge neutralization on abundant proteins (Munder *et al*, 2016). We speculated that this might also be associated with altered RNA interactions by cytoplasmic proteins.

Only around 10% of RNA-interacting proteins showed differences in crosslinking of greater than twofold, suggesting that the stress responses are relatively specific. This was supported by analyses of individual pathways of RNA metabolism. For example, ribosome biogenesis, which is expected to be inhibited by stress, appeared to be blocked at a specific step in the cytoplasmic maturation of the precursors to 60S subunits.

Multiple proteins involved in translation showed reduced interaction following sorbic acid treatment, presumably reflecting general translation inhibition. Less expectedly, these included factors reported to be P-body components: eIF4G (Tif4631 and Tif4632), the RNA helicase Ded1, and the poly(A) RNA binding proteins Pub1 and Pab1. Moreover, P-body components directly implicated in mRNA degradation also showed reduced RNA interactions, including the CCR4/NOT complex subunit Not3, decapping cofactors Pat1 and Edc3, surveillance factors, Upf3, Nmd2 and Ebs1, and the 5′ exonuclease Xrn1. In contrast, increased RNA interactions were observed for other P-body-associated proteins: These included the translation repressor Ssd1 and the dissociable RNAPII component Rpb7, as well as seven enzymes of intermediary metabolism, previously identified as P-body components. Increased RNA interaction upon sorbic acid exposure was also found for a further additional 29 proteins of intermediary metabolism, 7 of which were previously implicated as parts of P-bodies and stress granules. We speculate that stress signalling induced by sorbic acid exposure allows abundant proteins of intermediary metabolism to form more stable interactions with RNA, allowing PAR-TRAPP detection. The enhanced RNA interactions for intermediary metabolism enzymes with RNA could be a result of protein modifications accruing on these proteins under stress. Alternatively, the cytoplasmic acidification, proposed to be involved in stress signalling (Munder *et al*, 2016), could affect protein conformation, driving increased RNA binding. RNA–protein interactions can facilitate liquid–liquid phase separation into droplets (Lin *et al*, 2015), which build P-bodies, stress granules and other phase-separated regions. Thus, our expectation is that the relatively abundant proteins of intermediary metabolism could help stabilize and enhance the phase-separated regions arising under stress. In the short term, this may provide protection; however, prolonged stress would potentially allow droplets to mature into more filamentous states with negative consequences (Lin *et al*, 2015).

### Mapping precise amino acid–RNA contacts with iTRAPP

A previous report (Kramer *et al*, 2014) and our initial observations irradiation (preprint: Peil *et al*, 2018) indicated that adenosine, guanosine and cytosine are very poorly represented in RNA–protein crosslinks that can be identified by MS. The basis for this is unclear, since reciprocal CRAC and iCLIP mapping of interaction sites on the RNA shows no such extreme bias. However, we took advantage of this finding and developed the iTRAPP protocol based on 4tU labelling. This provides greater specificity for RNA binding proteins than 254 nm crosslinking, with reduced complexity of RNA crosslinks since only 4-thiouridine-containing species were expected.

Most amino acids can form photo-adducts with oligo-DNA or RNA sequences when single amino acids are added to the reaction (see e.g. Shetlar, 1980). In our analyses, crosslinks were identified with all amino acids, with over-representation of aromatic amino acids (Trp, Tyr, Phe) relative to their abundance in the crosslinked peptides, followed by positively charged amino acids (His, Lys, Arg). Less expectedly, the hydrophobic amino acids (Gly, Pro) were also enriched, possibly reflecting their role in unstructured domains (van der Lee *et al*, 2014).

Despite progress in the software to analyse RNA–peptide crosslinks, much remains to be done in this field. The chemistry of RNA–protein crosslinks is more complicated than just a sum of RNA and peptide masses. For example, hydrogen losses were demonstrated in the case of lysine crosslinking to 4-thiouridine (Kramer *et al*, 2011). The challenge is that allowing the flexibility required to identify all possible crosslinks and fragmentation products greatly increases the number of false-positive hits to a decoy database. Better understanding of RNA–protein crosslink chemistry would allow the development of software that is aware of specific predicted mass losses or gains depending on which amino acid participates in the crosslinking.

Comparison of precise sites of protein–RNA contact to published sites of protein modification revealed highly significant enrichment for proximity to reported sites of protein phosphorylation. $TiO_2$ enrichment we performed to isolate RNA-crosslinked peptides may introduce bias in favour of colocalization between protein phosphorylation sites and RNA crosslinks. As phosphorylation is unlikely to coexist with an RNA–protein crosslink (Kramer *et al*, 2014), we set up Xi to not consider phosphorylation as a possible protein modification. This leaves a small chance of identifying RNA crosslinked to a phosphorylated peptide as "RNA + phosphate" crosslinked to a peptide as masses of the former and the latter are identical. Careful discrimination between the two possibilities relies on the coverage of the potential phosphorylation site and RNA crosslink site in the MS2 spectra. Multiple ions, consistent with lack of phosphate at the potential phosphorylation site, are required to boost confidence in the "RNA+phosphate" hypothesis. Median fragment coverage of 70% achieved in our data suggests that in most of iTRAPP reported peptides there is evidence in favour of "RNA+phosphate" crosslink as opposed to RNA + phosphopeptide. Furthermore, 4-thiouridine diphosphate dominates both Lambda phosphatase-treated and Lambda phosphatase-untreated samples, indicating that the phosphate survives the phosphatase treatment, unlike the protein phosphorylation events, which were dramatically reduced.

We predict that protein–RNA association can be regulated via phosphorylation of RNA-interacting region causing electrostatic hindrance by negative charges. This could be defined by future analyses combining TRAPP with mutations in protein modifying enzymes.

The protocol described here can be readily adapted to other species. We anticipate that the resources provided here, a protocol to study protein–RNA binding sites at global scale and a list of protein–RNA attachment sites in *S. cerevisiae*, will stimulate systematic and detailed studies of the functionally important protein–RNA interface.

# Materials and Methods

## Reagents and Tools table

| Reagent/Resource | Reference or source | Identifier or catalog number |
|---|---|---|
| **Experimental models** | | |
| BY4741 derived *MATa, his3Δ1 leu2Δ0 met15Δ0 ura3Δ0, arg4Δ, Lys9Δ* | This study | |
| MG1655 derived *F-, Lambda-, rph-1 ΔlysA ΔargA* | Starosta *et al* (2014) | |
| **Chemicals, enzymes and other reagents** | | |
| Silica beads | Fluka | S5631-500g |
| Guanidine thiocyanate | Thermo Fisher Scientific | BP221-1 |
| RNaseA+T1 mix | Ambion | AM2286 |
| 4–20% Mini-PROTEAN TGX gel | Biorad | 4561095 |
| Imperial protein stain | Thermo Fisher Scientific | 24615 |
| DNase I | New England Biolabs | M0303 |
| Cyanase | Serva | 18542.01 |
| Nuclease P1 | Sigma | N8630 |
| Urea | Acros organics | 327380050 |
| Vivaspin polyethersulfone concentrator | GE Healthcare | 28-9323-61 |
| MagTrypsin | Takara | 635646 |
| TiO2 Mag Sepharose | GE healthcare | 28-9440-10 |
| Lambda phosphatase | New England Biolabs | P0753S |
| $[^{13}C_6]$-lysine | Cambridge Isotope Laboratories | CLM-2247 |
| $[^{13}C_6]$-arginine | Cambridge Isotope Laboratories | CLM-2265 |
| **Software** | | |
| *Xi v 1.6.731* | https://github.com/Rappsilber-Laboratory/XiSearch | |
| MaxQuant 1.6.1 | Cox and Mann (2008) | |
| Perseus 1.6.0.7 | Tyanova *et al* (2016) | |
| imputeLCMD package version 2.0 for R | https://rdrr.io/cran/imputeLCMD/ and Lazar *et al* (2016) | |
| **Other** | | |
| QExactive mass spectrometer | Thermo Fisher Scientific | |
| Fusion Lumos mass spectrometer | Thermo Fisher Scientific | |

## Methods and Protocols

### Yeast SILAC media and crosslinking

Yeast cells (BY4741 derived MATa, *his3Δ1 leu2Δ0 met15Δ0 ura3Δ0, arg4Δ, Lys9Δ*) were grown at 30°C under shaking in yeast nitrogen base media (Formedium CYN0410), containing 2% w/v of D-Glucose (Fisher G/0500/60), supplemented with Complete Supplement Mixture without tryptophan, lysine, arginine and uracil (Formedium DCS1339) and with 20 mg l$^{-1}$ uracil (Sigma U0750-100g). SILAC light media were additionally supplemented with 30 mg l$^{-1}$ lysine (Sigma L5626-100g) and 5 mg l$^{-1}$ arginine (A5131-100g), while 30 mg l$^{-1}$ 13C6 lysine (Cambridge Isotope Laboratories CLM-2247-H-0.25) and 5 mg l$^{-1}$ 13C6 arginine (Cambridge Isotope Laboratories CLM-2265-0.5) were added to SILAC heavy media. Equal cell mass (by OD measurement) of heavy-labelled and equal cell mass of light-labelled cells were combined per experiment.

For TRAPP using UVC (254 nm) crosslinking, the culture was harvested at OD$_{600}$ 0.5 from 750 ml of media and irradiated with Megatron apparatus (UVO3) as described (Granneman *et al*, 2011). Cells were harvested by filtration and immediately frozen for later processing.

For PAR-TRAPP, using UVA (350 nm) crosslinking, cells were grown to OD 0.15, 4-thiouracil (4tU) (Sigma 440736-1g) was added to 0.5 mM from 1 M DMSO stock solution, and cells were allowed to grow for 3 h. Cells were harvested by filtration and resuspended in 800 ml of growth media without 4tU and placed on the irradiation tray of the eBox irradiation apparatus. The UVA lamps were

pre-warmed for 1 min, and the cells were irradiated for 38 s delivering 7.33 J cm$^{-2}$ of UVA. Cells were harvested by filtration and frozen for later processing.

Equal cell mass of heavy-labelled and equal cell mass of light-labelled cells were combined per experiment before lysis step.

### E. coli SILAC media and crosslinking

*Escherichia coli* cells (MG1655 derived *F-, Lambda-, rph-1 ΔlysA ΔargA*) (Starosta *et al*, 2014) were grown in 150 ml of M9 minimal medium, supplemented with 1 μM FeNH$_4$SO$_4$, 0.2% w/v of D-Glucose (Fisher G/0500/60), 1 mM MgSO. SILAC light media were additionally supplemented with 40 mg l$^{-1}$ lysine (Sigma L5626-100g) and 40 mg l$^{-1}$ arginine (A5131-100g), while 40 mg l$^{-1}$ [$^{13}C_6$]-lysine (Cambridge Isotope Laboratories CLM-2247-H-0.25) and 40 mg l$^{-1}$ [$^{13}C_6$]-arginine (Cambridge Isotope Laboratories CLM-2265-0.5) were added to SILAC heavy media. Cells were grown to OD 0.5, diluted to 700 ml with growth media and irradiated with UVC (254 nm) using a Minitron (UVO3) (Granneman *et al*, 2011). Equal cell mass (70 ODs) of heavy-labelled and equal cell mass of light-labelled cells were combined per experiment before lysis step.

### Sorbic acid treatment

Yeast cells were treated as described above with 4tU in SILAC yeast nitrogen base media for 3 h. Sorbic acid was added to 6 mM final concentration from ethanol stock solution, and cells were incubated for 16 min. Cells were harvested and irradiated as described above.

### eBOX irradiation apparatus

The eBox is a chamber with internal dimensions of 49.5 × 25.5 × 48 cm (W × H × D) lined with aluminium foil. The sample tray made of borosilicate glass is positioned between 2 banks of 10 PL-L 36W/09/4P PUVA 350-nm bulbs (Philips 871150061410040). Each bank is driven by 5 PC236TCLPRO-TR ballasts (Tridonic 22176170). The two banks together consume 720 W of electrical power. UVA measurements were performed with Sper Scientific UVA/B Light Meter (850009), equipped with UVFS Reflective ND Filter, OD: 2.0 (THORLABS NDUV20B). Based on the spectral sensitivity of the meter, we estimate that in 38 s of irradiation time, we administer from 7.3 J cm$^{-2}$ of UVA (193 mW cm$^{-2}$). The UVA output of the eBox stabilizes after 1 min of lamp warmup. The sample is prevented from UVA exposure with sliding shutters during the warmup period, followed by shutters' extraction and sample irradiation for 38 s.

### Silica bead preparation

Silica beads (Fluka S5631-500g) were resuspended in 1 M HCl and left for 24 h at room temperature. The beads were then washed with water three times by spinning in a 50-ml falcon tube at 3,000 *g* for 2 min and decanting the supernatant, containing beads of the smallest size. After the last wash, beads are resuspended in volume of water to achieve 50% slurry suspension.

### Cell lysis

*Escherichia coli* cells pellets (140 ODs) were resuspended in 8 ml of 1:1 mix between phenol (Sigma P4557), saturated with 10 mM Tris–HCl pH 8.0, 1 mM EDTA and GTC buffer (4 M guanidine thiocyanate (Fisher BP221-1), 50 mM Tris–HCl pH 8.0 (Invitrogen 15504-020), 10 mM EDTA (Fisher D/0700/53), 1%

beta-mercaptoethanol), the sample was split between 8 2-ml screwcap tubes, 1 volume of zirconia beads (BioSpec 11079105z) was added, and cells were lysed with FastPrep-24 5 g (MP biomedicals 116005500) twice for 40 s at 6 m s$^{-1}$. Lysates were recovered into 50-ml falcon tube.

Yeast mixed cell pellets were resuspended in 2 ml of phenol-GTC buffer in a 50-ml falcon tube, 3 ml of zirconia beads was added, and cells were vortexed for 1 min 6 times, with 1-min incubation on ice between vortexing. 6 ml of phenol-GTC mix was added, and lysates were vortexed for additional 1 min.

### RNP isolation

- Spin down samples in 50-ml falcon tubes at 4,600 *g*, 4°C for 5 min.
- Recover the supernatant into 2-ml tubes and spin at 13,000 *g* for 10 min to clear the lysate.
- Recover the cleared lysate into a 50-ml falcon tube; avoid taking any pellet at this step.
- To the recovered lysate, add 0.1 volume of 3 M sodium acetate-acetic acid pH 4.0, mix briefly and slowly add 1 volume of absolute ethanol, vortex to mix.
- Add 1 ml of 50% slurry suspension of silica in water (per 400 ODs of yeast cells), followed by 0.5 ml of ethanol to maintain 50% ethanol concentration in the mix.
- Load nucleic acids onto silica for 60 min on a rotating wheel at room temperature.
- Spin down the beads at 2,500 *g*, 4°C for 2 min and resuspend the beads by vigorous vortexing in 15 ml of wash buffer I (4 M guanidine thiocyanate, 1 M sodium acetate (Alfa Aesar 11554)–acetic acid (Fisher A/0400/PB17) pH 4.0, 30% ethanol). At this point, samples can tolerate harsh mixing; therefore, the use of a FastPrep or a similar homogenizer with the corresponding 50 ml adapter is possible.
- Pellet the resuspended silica at 2,500 *g*, 4°C for 2 min, and discard the supernatant. One can use a vacuum system or pipette to remove as much of wash buffer as possible. Repeat the wash with buffer I 2 more times.
- Wash the beads three times with 20 ml of wash buffer II (100 mM NaCl, 50 mM Tris–HCl pH 6.4, 80% ethanol) using the procedure described for wash buffer I.
- After the third wash, resuspend the beads in a small amount of wash buffer II using a vortex or a homogenizer. Transfer the resuspended silica to two 2-ml tubes.
- Spin the tubes at 2,000 *g* for 2 min at room temperature and discard the supernatant.
- Dry the silica for 10 min in a SpeedVac at 45°C.
- Elute the nucleic acids three times with 500 μl per 2-ml tube. Use vortex or a homogenizer to resuspend the silica and spin at 4,000 *g* for 1 min to pellet the silica. Recover the supernatant into two new 2-ml tubes. As the dissolved nucleic acids make the recovered supernatant viscous, some silica beads will be recovered together with the supernatant.
- Spin the recovered 2 × 1.5 ml supernatant at 14,000 *g* for 1 min at room temperature. Transfer the supernatant into 2 fresh 2-ml tubes and repeat the spin.
- Recover the supernatant into two 2-ml protein LoBind tubes (Eppendorf 0030108132). Add 0.5 μl of RNaseA + T1 (Ambion AM2286) mix per tube and incubate the samples for 2 h in a

SpeedVac at 45°C, followed by SpeedVac incubation at 65°C until the samples are fully dry. Alternatively, the sample can be left overnight in a SpeedVac at 45°C, which takes approximately 16 h to dry. If the sample is expected to have a large amount of DNA, consider adding a DNAse step or using a promiscuous nuclease such as benzonase or cyanase.

- The dried samples are resuspended in 75 μl Laemmli buffer per 2-ml tube by vortexing, incubated at 100°C for 10 min. At this point, the two 2-ml tubes are merged into one.

- Resolve approximately 10 μl on a gradient 4–20% Mini-PROTEAN TGX gel (Biorad 4561095), and stain with Imperial protein stain (Thermo 24615) according to the manufacturer's protocol. The amount of sample to be loaded on a gel depends on the concentration of proteins in the sample. A good strategy is to run a test gel with a few varying loading to determine the optimal amount of sample to process.

- Each lane of the gel, corresponding to one SILAC mix was cut into several gel fractions. The extent of fractionation depends on the mass spectrometer. In our hands, even a single fraction can give up to 1,000 quantified proteins; however, this number grows with the number of fractions with diminishing returns.

- De-stain the excised fractions and digest the proteins with trypsin, as previously described (Shevchenko et al, 1997). Briefly, reduce the proteins in 10 mM dithiothreitol (Sigma-Aldrich, UK) for 30 min at 37°C and alkylate in 55 mM iodoacetamide (Sigma-Aldrich, UK) for 20 min at ambient temperature in the dark. Finally, digest the proteins overnight at 37°C with 12.5 ng $\mu l^{-1}$ trypsin (Pierce, UK).

Following digestion, samples were diluted with equal volume of 0.1% TFA and spun onto StageTips as described (Rappsilber et al, 2003). Peptides were eluted in 20 μl of 80% acetonitrile (ACN) in 0.1% trifluoroacetic acid (TFA) and concentrated down to 4 μl by vacuum centrifugation (Concentrator 5301, Eppendorf, UK). The peptide sample was then prepared for LC-MS/MS analysis by diluting it to 5 μl by 0.1% TFA.

Where indicated, treatment of nucleic acids with DNase was performed in buffer containing Tris pH 8.0, 2.5 mM MgCl$_2$, 0.5 mM CaCl$_2$, using DNase I from New England Biolabs (M0303) at 100 U mg$^{-1}$ of DNA, supplemented with 200 U of RNAsin (Promega N211A). Treatment with Cyanase (Serva 18542.01) was performed in Tris pH 8.0 buffer supplemented with 6 mM MnSO$_4$, using 0.1 U of enzyme per 1 μg of nucleic acids. Silver staining was performed as previously described in (Yan et al, 2000).

### iTRAPP

Yeast cells were irradiated with UVA as described above. Cultures of 10 l or 2.5 l of cells were processed as one sample. Cells were lysed, and RNA-associated protein was isolated as described above for PAR-TRAPP. Following RNA elution from silica beads, sodium acetate pH 5.3 was added to 10 mM, ZnCl$_2$ to 0.5 mM and DTT to 0.5 mM final concentrations. Nuclease P1 (Sigma N8630) was added 1:4,000 weight of RNA:weight of enzyme, and the sample was incubated at 37°C overnight to completely degrade bound RNA to nucleotide 5′ monophosphate products. Tris–HCl pH 8.0 buffer was added to 50 mM, DTT was added to 30 mM, and solid urea (Acros Organics 327380050) was added for 8 M final concentration to reduce and denature proteins. The solution was passed through a 30KDa MWCO Vivaspin Polyethersulfone concentrator (GE

Healthcare 28-9323-61), and flow through was discarded. Denaturation at this step is essential to prevent proteins smaller than 30 kDa from passing through the membrane (Wiśniewski et al, 2011). Retentate was washed three times with urea buffer (8 M urea in 50 mM Tris–HCl pH 8.0), and cysteines were blocked by passing through a solution of 50 mM IAA in urea buffer. The retentate was washed three times with 2 mM DTT in urea buffer to quench the IAA. Finally, retentate was recovered, urea was diluted to 1 M with 50 mM Tris pH 8.0. This process removed of 99.4% of the nucleic acids initially present in the sample (~40 mg).

The sample was added to trypsin-conjugated magnetic beads (Takara 635646) and incubated for 12 h with mixing at 37°C. The trypsin beads were removed from the supernatant, and the sample was diluted with water 1:1. Sodium acetate was added to 10 mM, ZnCl$_2$ to 0.1 mM, and pH was reduced to approximately 5.2 with acetic acid. To ensure complete RNA degradation, a further 20 μg of nuclease P1 was added and the sample was digested at 37°C for 24 h. After this digestion, the sample was either left untreated or incubated with Lambda phosphatase (λ). The pH was brought to 8.0 with 1 M Tris–HCl, and DTT was added to 2 mM, EDTA pH 8.0 to 1 mM and MnCl$_2$ to 2 mM final concentration. To the "+phosphatase sample", 800 units of λ phosphatase (NEB P0753S) was added and incubated for 1 h at 30°C, while the "−phosphatase sample" was mock treated.

TFA was added to 1%, ACN to 2.5% final concentration. Sample was loaded onto 4 C18 SPE cartridges (Empore 4215SD). Washed with 0.1% TFA and eluted in 0.1% TFA, 80% ACN. Eluates were dried in SpeedVac. Next, the sample was resuspended in 250 μl of binding buffer (50% ACN, 2 M lactic acid) with vortexing and brief sonication. Phospho-group containing RNA-crosslinked peptides were the captured with TiO$_2$ Mag Sepharose beads (GE healthcare 28-9440-10) according to the manufacturer's protocol. Peptides eluted from TiO$_2$ sepharose were loaded onto C18-StageTips as described above followed by LC-MS analysis.

### LC-MS analysis

LC-MS analyses were performed on an Orbitrap Fusion Lumos Tribrid Mass Spectrometer (Thermo Fisher Scientific, UK) coupled online, to an Ultimate 3000 RSLCnano Systems (Dionex, Thermo Fisher Scientific, UK). Peptides were separated on a 50 cm EASY-Spray column (Thermo Scientific, UK), which was assembled on an EASY-Spray source (Thermo Scientific, UK) and operated at 50°C. Mobile phase A consisted of 0.1% formic acid in LC-MS grade water, and mobile phase B consisted of 80% acetonitrile and 0.1% formic acid. Peptides were loaded onto the column at a flow rate of 0.3 μl min$^{-1}$ and eluted at a flow rate of 0.2 μl min$^{-1}$ according to the following gradient: 2–40% mobile phase B in 136 min and then to 95% in 11 min. Mobile phase B was retained at 95% for 5 min and returned back to 2% a minute after until the end of the run (160 min). The iTRAPP analysis was performed on a QExactive mass spectrometer (Thermo Fisher Scientific, UK). The same LC conditions applied as described above.

For orbitrap Fusion Lumos, FTMS spectra were recorded at 120,000 resolution (scan range 400–1,900 $m/z$) with an ion target of 4.0e5. MS2 in the ion trap at normal scan rates with ion target of 1.0E4 and HCD fragmentation (Olsen et al, 2007) with normalized collision energy of 27. The isolation window in the quadrupole was 1.4 Thomson. Only ions with charge between 2 and 7 were selected for MS2.

For QExactive, FTMS spectra were recorded at 70,000 resolution and the top 10 most abundant peaks with charge $\geq 2$ and isolation window of 2.0 Thomson were selected and fragmented by higher-energy collisional dissociation with normalized collision energy of 27. The maximum ion injection time for the MS and MS2 scans was set to 20 and 60 ms, respectively, and the AGC target was set to 1 E6 for the MS scan and to 5 E4 for the MS2 scan. Dynamic exclusion was set to 60s.

The MaxQuant software platform (Cox & Mann, 2008) version 1.6.1.0 was used to process the raw files, and search was conducted against the *S. cerevisiae* reference proteome set of UniProt database (released on November 2017), using the Andromeda search engine (Cox *et al*, 2011). The first search peptide tolerance was set to 20 ppm while the main search peptide tolerance was set to 4.5 ppm. Isotope mass tolerance was 2 ppm and maximum charge to 7. A maximum of two missed cleavages were allowed. Carbamidomethylation of cysteine was set as fixed modification. Oxidation of methionine, acetylation of the N-terminal and phosphorylation of serine, threonine and tyrosine were set as variable modifications. Multiplicity was set to 2, and for heavy labels, arginine 6 and lysine 6 were selected. Peptide and protein identifications were filtered to 1% FDR. For the iTRAPP experiment, peak lists were generated from MaxQuant version 1.6.1.0. The Xi software platform was used to identify proteins crosslinked to RNA.

### TRAPP UV enrichment data analysis

Peptides.txt file, generated by MaxQuant software, was used for the analysis. First, peptides mapping to contaminants and decoy database were removed. The reported heavy and light intensities per peptide were exported into Rstudio version 1.1.453 (http://www.rstudio.com/). Missing peptide intensity values were imputed using impute.minprob function of the imputeLCMD package version 2.0 (Lazar *et al*, 2016) (https://rdrr.io/cran/imputeLCMD/). For *S. cerevisiae* PAR-TRAPP and *E. coli* TRAPP data, we run impute.minprob function with the following parameters: $q = 0.1$ tune.sigma = 0.01. For *S. cerevisiae* TRAPP data, tune.sigma parameter was set to 0.0035. If in a given biological replicate a peptide was missing intensity in both + UV and −UV, the imputed values were removed. We then calculated +UV to −UV protein ratio in each experiment as median of peptides using "leading razor protein" as protein identifier. Proteins identified with at least two peptides in at least two experiments were considered for further analysis. The protein ratios were $\log_2$ transformed, and the statistical significance of the result was determined using the Limma package version 3.36.2 (Ritchie *et al*, 2015).

### iTRAPP data analysis

The targeted Xi search was performed with cleavable RNA modifications limited to up to trinucleotide RNA, where at least one residue is 4tU (Dataset EV6). The targeted masses can be identified on top of the otherwise defined variable modifications. Note that only one of the specified masses can be identified per PSM to prevent very long search times. Since the crosslink was assumed to always happen through the 4tU residue, the masses allowed to remain on fragment peptide ions were derived from 4tU: 128.004435 (4tU base); 94.016713 (4tU base-H2S); 226.058971 (4-thiouridine monophosphate-HPO3-H2S); 306.025302 (4-thiouridine

monophosphate-H2S); 340.013027 (4-thiouridine monophosphate); 260.046696 (4-thiouridine monophosphate-HPO3); 385.991636 (4-thiouridine monophosphate+HPO3-H2S); 419.979358 (uridine monophosphate+HPO3).

FDR was calculated on a list of peptide spectral matches, sorted by descending match score, as a ratio between the number of decoy hits and the number of target hits for a given score.

Xi search set-up: The raw data from QExactive mass spectrometer were processed into peak lists using MaxQuant 1.6.1.0. We then performed a search using Xi version 1.6.731 (https://github.com/Rappsilber-Laboratory/XiSearch) (preprint: Mendes *et al*, 2018) search against the reference proteome with the following search parameters:

MS1 tolerance 6 ppm; MS2 tolerance 20 ppm; enzyme—trypsin \p; missed cleavages allowed—2; minimum peptide length—6; fixed modifications—carboxymethyl at cysteine; variable modifications—oxidized methionine; maximum number of neutral losses per MS2 fragment—1; report MS2 fragments that are off by 1 Da—true; complete search configuration can be found in the Dataset EV10.

RNP[xl] plugin: We utilized RNP[xl] plugin (Veit *et al*, 2016) for Proteome discoverer version 2.1.1.21 (Thermo Fisher Scientific). MS1 tolerance 6 ppm; MS2 tolerance 20 ppm; enzyme—trypsin \p; missed cleavages allowed—2; maximum RNA length—3 nucleotides of which one has to be 4tU. RNA modifications—HPO3-H2S; −H2S; +HPO3-H2S; +HPO3; −HPO3; fixed modifications—carboxymethyl at cysteine; variable modifications—oxidized methionine;

In order to be able to calculate FDR for RNP[xl] hits, we constructed a combined target + decoy database using Decoy database builder software (Reidegeld *et al*, 2008). The software was run in combined mode (reverse + shuffled + random decoy database). FDR was calculated on a list of peptide spectral matches, sorted by descending score, as a ratio between the number of decoy hits and the number of target hits for a given score. Peptide to spectrum matches passing the 1% FDR was filtered to remove PSMs with undefined crosslink site.

The data for both RNP[xl] and Xi were further processed in MS Excel to remove duplicates based on protein and crosslinked amino acid to avoid over-representation of individual highly abundant crosslink sites, observed in multiple PSMs. The list was further filtered to remove duplicates with same peptide and same RNA to reduce over-representation of abundant peptides. Each filtering step was performed so as to keep peptide to spectrum matches with highest match score.

### Scoring for colocalization of RNA–protein crosslinks with sites of post-translational protein modifications

The modification sites for proteins recovered in iTRAPP were obtained from UniProt database on 12.07.2018 (The UniProt Consortium, 2017). We then calculated the number of observed crosslinks within 20 amino acids from a phosphosite (X-Pi[observed]) and the number of observed crosslinks outside the 20 residue window (X[observed]). In order to obtain the expected values, we shuffled the phosphosites using the MS excel "RANDBETWEEN" function in the crosslinked proteins and calculated the (X-Pi[expected]) and (X[expected]). We performed the shuffling 100 times and calculated the average (X-Pi[expected]) and (X[expected]) values, Finally, we performed a chi-square test to identify if the observed values were significantly different from the averaged expected values.

### Bioinformatics resources

We utilized The Yeast Metabolome Database (Ramirez-Gaona et al, 2017) and The E. coli Metabolome Database (Sajed et al, 2016) to identify proteins involved in intermediary metabolism. The analysis of enrichment of proteins belonging to the GO terms in Figs 1B and C, 2B, 3B and C, 4B and C, and 5B was performed with Perseus software (Tyanova et al, 2016). GO term annotations for the analysis were downloaded from annotations.perseus-framework.org on the 29.11.2017 for S. cerevisiae and on 25.02.2018 for E. coli. The analysis shown in Figs 1E and F, and 4F, Appendix Figs S1A and B, and S2 was performed using David 6.8 web service (Jiao et al, 2012). Venn diagrams were generated using the BioVenn web service (Hulsen et al, 2008). To identify orthologous clusters between E. coli and S. cerevisiae, we took advantage of the Inparanoid 8 database (Sonnhammer & Östlund, 2015). Protein abundance data were obtained from the PaxDb database (Wang et al, 2015). Integrated dataset for S. cerevisiae was used for the analysis. The ratio between enriched and total proteins in Figs 2D and 4E was calculated by dividing the number of proteins in the bin annotated as UV enriched divided by the number of enriched proteins + the number of non-enriched proteins. The non-enriched proteins are defined as proteins quantified with 2 or more peptides in at least two experiments, but which fail to score a statistically significant enrichment upon UV irradiation (adjusted $P$-value higher than 0.05).

### Sequencing data analysis

#### Pre-processing

Raw datasets were demultiplexed according to barcode using a custom python script. Flexbar (Dodt et al, 2012) was used to remove adaptor sequence, trim low-quality bases from the 3′ end, and remove reads with low-quality scores (parameters –u 3 –at 1 –ao 4 –q 30 with adapter sequence TGGAATTCTCGGGTGC CAAGGC). In addition to the barcode, each read contained three random nucleotides to allow PCR duplicates to be removed by collapsing identical sequences with fastx_collapser.

#### Alignment

Mapping sequencing reads to tRNAs is complicated by the fact that most tRNAs are present in multiple identical copies throughout the genome. Furthermore, mature tRNAs differ substantially from the genomic sequence: all tRNAs receive a CCA trinucleotide at the 3′ end, and some transcripts are post-transcriptionally spliced. For these reasons, sequencing reads were aligned to a database consisting of only mature tRNA sequences. A fasta file containing mature tRNA sequences was downloaded from Ensembl, and identical tRNAs were collapsed into single genes using fastx_collapser. Subsequently, the tRNA genes were concatenated into a single "chromosome" with 40 base pair spacers between each tRNA using a custom python script. Similar files were also made for mRNAs and ribosomal RNAs. CRAC reads were mapped against this artificial genome using Novoalign (V2.07.00, Novocraft) and parameters –s 1 –l 17 –r None (to ensure only uniquely mapping reads). The accompanying Novoindex file was generated using parameters –k 10 –s 1. To remove PCR duplicates that were not previously discarded during pre-processing because of sequencing errors or differential trimming

at the 3′ end, any reads with the same random nucleotides and with 5′ ends mapping to the same coordinates were collapsed into a single read (Tuck & Tollervey, 2013).

#### Downstream analysis

Downstream analyses were performed using the pyCRAC software (Webb et al, 2014). To count overlap with features, pyReadCounters (pyCRAC) was used together with a custom annotation file corresponding to the concatenated genome. Plots showing binding across tRNAs (Fig 7C and D) were generated with pyGTF2bedGraph (pyCRAC) using the results from pyReadCounters as input. The coverage at each position along the genome was normalized to the library size using the –permillion option. Putative crosslinking sites (deletions) were mapped using the –t deletions option. The heatmaps in Fig 7E were generated using the –outputall option in pyBinCollector and then plotted in Excel.

## Data availability

The datasets produced in this study are available in the following databases:

- All sequence data: Gene Expression Omnibus accession number GSE119867 (https://www.ncbi.nlm.nih.gov/geo/query/acc.cgi?acc=GSE119867).
- Mass spectrometry proteomics data have been deposited to the ProteomeXchange Consortium via the PRIDE (Vizcaíno et al, 2016) partner repository with the dataset identifier PXD011071 (https://www.ebi.ac.uk/pride/archive/projects/PXD011071).

Expanded View for this article is available online.

## Acknowledgements

VS was funded by the Swiss National Science Foundation (P2EZP3_159110). The Wellcome Trust generously funded this work through a Principal Research Fellowship to D.T. (077248), a Senior Research Fellowship to J.R. (103139) and an instrument grant (108504), with additional support from the Wellcome Trust/Edinburgh University via the Institutional Strategic Support Fund. Work in the Wellcome Centre for Cell Biology is supported by a Centre Core grant (203149).

## Author contributions

VS and DT designed research; VS and EP performed biochemistry; CS and VS and performed and analysed MS data, CS, VS and LF performed bioinformatics; and VS, SB, CS, JR and DT analysed data and wrote the paper.

## Conflict of interest

The authors declare that they have no conflict of interest.

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
