## [Review Process File · Molecular Systems Biology]

Defining the RNA Interactome by Total RNA-Associated Protein Purification

Vadim Shchepachev, Stefan Bresson, Christos Spanos, Elisabeth Petfalski, Lutz Fischer, Juri Rappsilber and David Tollervey.

Review timeline:

Submission date:	16 th October 2018
Editorial Decision:	7 th November 2018
Revision received:	1 st February 2019
Editorial Decision:	25 th February 2019
Revision received:	5 th March 2019
Accepted:	13 th March 2019

Editor: Maria Polychronidou

Transaction Report:

1st Editorial Decision

7th November 2018

Thank you again for submitting your work to Molecular Systems Biology. We have now heard back from the two referees who agreed to evaluate your study. As you will see below, the reviewers acknowledge that the presented method for identifying RNA-binding proteins is a potentially useful contribution to the field. They raise however a series of concerns, which we would ask you to address in a major revision.

Without repeating all the points listed below, in particular reviewer #2 raises substantial issues related to the methodology, which need to be convincingly addressed.

All other issues raised by the reviewers need to be satisfactorily addressed. As you may already know, our editorial policy allows in principle a single round of major revision so it is essential to provide responses to the reviewers' comments that are as complete as possible. Please feel free to contact me in case you would like to discuss in further detail any of the issues raised by the reviewers.

REFeree REPORTS.

Reviewer #1:

Shchepachev et al. describe TRAPP, a new method for the discovery of RNA-binding proteins (RBPs) via silica-based enrichment of crosslinkend RNA-protein complexes. The method is validated by RNase/DNase treatment controls and applied to yeast and *E. coli*, including a stress condition in yeast. PAR-TRAPP is further introduced as a modified version that utilizes 4tU in order to refine the method, whereas iTRAPP allows the determination of RNA interaction sites on proteins. Finally, the authors pick yeast enolase 1, predicted to be an RBP by their assay, for downstream validation by CRAC, and show that this protein mostly binds cytoplasmic tRNAs.

The method is described in sufficient technical detail and will be of great value for the field. Researchers could apply it even without MS analysis for a protein of interest, detecting that protein by, e.g., immunoblot. However, we recommend several issues to be addressed prior to publication.

Abstract

How does the discovery of Enolase interaction with tRNAs fit to the sentence in the results part: 'Eno2, was previously reported to interact with mitochondrial tRNAs (Entelis, Brandina et al., 2006)'. If the discovery is limited to Eno1, this should be stated here.

Introduction

p. 3, 1st paragraph: References would be helpful after the sentence: "Both approaches have been used extensively in many systems."

p. 3, 2nd paragraph: Double-check whether the references (Garcia-Moreno, Noerenberg et al., 2018, Gatto, Breckels et al., 2018, Trendel, Schwarzl et al., 2018) really fit with the extraction methods described.

p. 3, 3rd paragraph: The authors jump into listing the benefits of TRAPP techniques before describing what these are.

The authors claim that TRAPP can reveal the 'steady-state protein-RNA interactome'. This may be true for a relative comparison between conditions. Nevertheless, a precise determination of steady-state conditions requires a truly quantitative, unbiased crosslinking approach which is not the case here. The authors should revise this statement.

Results

p. 5, 2nd paragraph: The authors discuss the specificity for RNA by the UV crosslinking efficiency. In addition to this, an important step in the protocol is the ethanol-based wash in TRAPP which washes away bound DNA (Avison M (2008) Measuring Gene Expression. London, UK: Taylor & Francis). The authors should add this point.

p. 5, 2nd paragraph: Could the authors please elaborate more on the quantity of 'some proteins'. It would be helpful to see a GO-term based upon absolute quantification of RBPs, DNA-binding, and other proteins for validation of the method based on total lysate MS. In comparison to the total proteome, in TRAPP/PAR-TRAPP/iTRAPP the RBPs should be enriched. The enrichment could be taken into account by binning and the expected result would be an increased ratio of RBPs in bins with higher enrichment. In addition, the enrichment levels should be correlated to the total protein levels.

p. 5, last paragraph: "Several reports ..." needs a reference.

p. 6, last paragraph should be rephrased to clarity, it is hard to follow.

p. 7, 1st paragraph: The authors postulate that abundant proteins have acquired features that favour transient RNA interaction. This should be referenced or elaborated in detail. How is this different from the well-known unspecific crosslinking of RNA with abundant proteins at high concentration?
p.7: The authors should use percentage indications rather than absolute numbers, because the protein levels recovered between TRAPP and PAR-TRAPP are very different.

p. 7, 2nd paragraph: The authors should clarify the term 'dynamic range', and discuss the values below $\log_2 = 1$ compared to the pA data in Fig. 2B,C.

p. 7, 3rd paragraph: The authors present a valuable data set for bacterial candidate RBPs but this area is neither properly introduced and referenced nor discussed in sufficient detail. Which RBPs would one expect to detect in E. coli or related bacteria, which of them were detected in the end? Are copy numbers known? This is the more important since the entire next paragraph is then devoted to proteins that have well-established primary functions unrelated to RNA-binding, so regardless of their detection here, one would rather consider them contaminants. For the proteins listed in the last sentence, enrichment factors and references are missing.

p. 8, 1st paragraph: The authors refer to a similar correlation between protein abundance and TRAPP enrichment in yeast and E. coli. Unfortunately, the bacterial data is not shown. A suppl. figure correlating protein abundance (MS of lysate input) and enrichment would be necessary for yeast and E. coli in order to draw valid conclusions here.

p. 10, 3rd paragraph and Fig. 4A: Only the differences in TRAPP enrichment are compared, but how does the protein level change upon stress? The authors should compare the lysate protein abundance between the stressed and non-stressed conditions to make sure that the enrichment they observe is not due to differing protein levels.

p. 8, 1st paragraph: Fig4B should be 3B.

p. 10, 1st paragraph: The authors propose an effect of acid stress on early ribosomal biogenesis.

How do the authors exclude an effect in translation that is propagated to ribosome biogenesis by a lack in the production of biogenesis factors? The authors should check for equal levels of biogenesis factors in both conditions to bolster their claim.

p. 20: The authors should clarify how the SILAC based enrichment was determined. Were heavy and light samples been mixed prior to x-linking or was only the heavy sample x-linked? The text does not clearly explain this, whereas in Fig1A it seems like the ¹³C-labeled cells were x-linked and the ¹²C-labeled cells were not.

Other comments

p. 5, 2nd paragraph: The authors should rephrase this sentence: "Degradation of DNA had little effect on protein recovery, whereas this was substantially reduced by RNase treatment". It is not clear at which stage of TRAPP the protein recovery refers to.

p. 5, 3rd paragraph: The sentence about the 'missing' peptides should be rephrased.

p. 12, 2nd paragraph: The second PO4 should be indicated in the brackets.

p. 13 and 14, Fig. 7 should be 6.

Fig. 1E: The top row should be a heading.

Fig. 1D: The green ring must be bigger to match the proportion of proteins. Add a circle for YMDB proteins that are compared in the text.

Fig. S2A: RNase and DNase should have a lower case 'a'.

Fig. 3C,D: How are conserved RBPs and non-RBPs defined? Fig. 3D should be referred to in the main text. It would be important to also give GO-term processes for known RNA-binding proteins as a positive control to the newly identified ones.

Reviewer #2:

In this manuscript, Shchepachev et al introduce a generic method for the isolation and characterization of RNA-associated proteins. They achieved this by combining UV-crosslinking, silica-mediated isolation of RNA-protein conjugates, and mass spectrometry for protein identification. This method, termed TRAPP, is accompanied by a similar methodology where nascent RNA is labeled with 4-SU, followed by crosslinking at slightly higher wavelength, and further work-up as for TRAPP. The authors found TRAPP to be more sensitive (i.e. identifying more proteins), and they identify and partially confirm RNA-binding of novel proteins. Furthermore, they determined global changes in RNA-binding of acid-stressed cells, and identified RNA-crosslinking sites in many proteins in a procedure termed iTRAPP, which includes a TiO₂ enrichment step to isolate RNA-crosslinked peptides.

This manuscript follows a series of studies that have appeared over the last few years to characterize proteins that interact with poly-adenylated RNA, however now extending this to (non-coding) RNAs lacking a poly(A) tail. This is highly relevant given the often poorly characterized mechanisms that mediate functionality of RNA. However, the manuscript suffers from major shortcomings with regard to the validity and rigor of the used methodology, casting doubt on some of the chief findings of the paper.

Major concerns:

1. The main concern about the TRAPP methodology, and primarily the used conditions for UV-irradiation, is that the data strongly indicate over-crosslinking. In particular, the observation that protein crosslinking scales with protein abundance (Figure 1g) is alarming: the authors should have recognized this as an emblematic indication of over-crosslinking, and thus of non-specific RNA-protein interactions, instead of postulating that 'abundant proteins have features that predispose them towards RNA association' (page 6), that 'highly abundant proteins have systematically acquired features that favor transient RNA interaction' (page 7), or, as a quite astonishing claim that there is no specificity after all in the way RNA interacts with protein, but that instead RNA and protein 'rather act biophysically "en masse"' (page 16). Indeed they are crosslinked 'en masse' - probably as an artifact.

In fact the authors themselves provide additional evidence for over-crosslinking in TRAPP, by showing that the correlation between protein abundance and crosslinking to RNA does not exist in PAR-TRAPP, at the same time identifying half the number of proteins (Fig 1g): using 4-SU as the

crosslinking base in PAR-TRAPP will exhaust crosslinking as soon as all 4-SU are photoactivated. In TRAPP, this seems not to be the case with 254 nm crosslinking where one simply starts crosslinking everything in the surrounding of RNA. In a way the authors propose this as a potential mechanism (page 15), without recognizing it as a pitfall. Over-crosslinking is likely the result of UV radiation at higher intensity than commonly used for RNA-interactome analysis, in the newly-constructed device (page 4-5, Fig S1E). Although mathematically cells may have received the same cumulative dose in both cases, I could very well imagine that artifacts may be introduced when irradiating cells at high intensity. In fact, the absence of a correlation between RNA-interaction and protein abundance in previous RNA-interactome studies has been a strong argument for the specificity of the observed interactions.

Collectively, as a result of this, it is highly questionable how the results obtained by TRAPP (Fig 1, 3 and all associated supplemental material) can be trusted. To remedy this fundamental issue, the authors should titrate the UV radiation dose, monitor when high abundance proteins start to disappear, and then accordingly adjust the radiation intensity and duration to minimize background caused by non-specific interactions.

2. The iTRAPP workflow includes a TiO₂ enrichment step to purify RNA-crosslinked peptides (Fig 5). Surprisingly, only 48% of the observed crosslinking sites fall within known RNA-binding domains, which contrasts with previous studies who find >85% (e.g. Kramer et al). In addition, the authors find that RNA-protein crosslinks strongly correlate with sites of phosphorylation (Figure 5e). Both observations may be explained by the likelihood that in fact TiO₂ strongly enriched for phospho-peptides (which can be easily tested by including phosphorylation as a PTM in the database search). This would not be surprising at all if the authors omitted a phosphatase step, meaning that inclusion of such a treatment should result in a distinct improvement of the results.

Other concerns:

3. TRAPP critically relies on silica enrichment of RNA-crosslinked proteins. Yet, proteins or RNA-crosslinked proteins are usually not retained by silica - do the authors have a proposed mechanism how this may still occur? In particular, how would a (large) protein crosslinked to a (small) RNA be retained, while the same free protein would not?
4. Fig 2b and c raise a couple of questions: i) the figure is hard to understand without declaring Si and pA. ii) the difference between panels b and c is unclear. iii) Authors claim that GO terms like RNA binding, translation etc have a higher dynamic range (page 7). This is not true, instead they have a larger spread in their enrichment. iv) It is unclear how black dots in TRAPP (i.e. not passing significance) are still classified as TRAPP-enriched. Similarly, some colored dots are apparently significant, although showing a fold change <1, or some even <0 (i.e. depleted?). v) in general, the scatter plots as shown in fig 2bc and elsewhere in the paper are only partially informative since they only show ratios, not significance. Therefore they should be replaced by volcano plots to indicate both parameters.
5. Figure 3a: why are so many proteins identified in the negative control (-UV), or even enriched there (e.g. membrane proteins)?
6. In their analysis of dynamic changes in RNA-protein interactions (page 8) the authors prefer to use PAR-TRAPP over TRAPP. The reasoning is a bit surprising after their prior claims that TRAPP outperforms PAR-TRAPP. However, the lack of dynamic range in the observed stress-induced interaction differences in TRAPP is fully understandable in the light of over-crosslinking: all RNA-protein crosslinks are saturated, both in the absence and presence of stress, hence levelling out any differences that may exist.
7. It is unclear why the authors propose a defect in pre-60S ribosomes (page 10), and not a functional change?
8. For iTRAPP, 10 liters of yeast culture was used. Since this is a very large amount, it will be useful if the authors can comment why they need this large volume (especially compared to other methods), or at which steps (Fig 5a) the major losses are experienced.
9. The discussion contains a long section on RNA-mediated phase separation. Although this is a popular topic, the section is highly speculative, and none of what is discussed is based on data produced or shown in the paper. It should thus be eliminated.
10. Page 12: it is unclear how the authors derive crosslinking efficiency for each amino acid. Specifically, in many peptides (Table S8) most if not all amino acids score very low (<0.1) for localisation of crosslinks, i.e. I assume that this means that no crosslink site could be determined.

First question is why this is, second how this is handled in data interpretation, i.e. on which basis were peptides in/excluded for scoring as in Fig 5C. For instance, if a W in a 15-amino acid peptide scores 0.2 for its crosslinking likelihood (table S8), is it considered crosslinked or not? And how is the value (0.2) propagated to calculate a 'fold change' (Fig 5c)?

11. On page 17 the authors say: 'Only around 10% of RNA-interacting proteins showed differences in crosslinking of greater than 2-fold, suggesting that the stress responses are relatively specific'. I cannot follow the logic of this - instead I would interpret the observation that the response is mild, not specific.

12. On page 18 the authors say: 'We speculate that stress signaling allows abundant proteins of intermediary metabolism to form more stable interactions with RNA, allowing PAR-TRAPP detection'. It is unclear what this speculation is based on, and actually what it means: How can protein abundance, as a biophysical property, induce formation of 'more stable' interactions?

13. In several instances the authors draw far-reaching conclusions where more simple explanations seem more obvious. For instance: i) Page 13: Eno1 was identified to interact with mRNA encoding Eno2. Since these are among the most abundant protein and RNA, respectively, in a cell, this strongly suggests a coincidental interaction. ii) In addition, the authors find that Eno1 interacts with cytoplasmic and not mitochondrial tRNAs, leading them to conclude that 'these reflect bona fide, in vivo interactions' (page 14). An easier and more likely interpretation is that, as is known, Eno1 is localized in the cytoplasm and not in mitochondria. iii) Eno1 was found to interact with the T-loop, anti-codon or extended variable loop in tRNA (page 14), leading the authors to conclude that 'Eno1 binds to specific structural elements'. Instead, it is more likely that these are the only sites in a tRNA that can crosslink because the other ones are engaged in base-pairing in the tRNA itself. This may be verified in a CLIP experiment (under conventional crosslinking conditions) for an amino-acyl-tRNA synthetase. To conclude this, the authors say that 'There is every reason to think that these data capture genuine, in vivo RNA-protein contacts'. Apart from the question if stylistically this is a fitting sentence, I wonder if it is actually true.

1st Revision - authors' response

1st February 2019

We have responded to all of the points raised by the referees and have included a substantial amount of additional data. Notably, the TRAPP analyses on yeast and E.coli have both been repeated in triplicate using two additional UV doses; the iTRAPP analyses have been repeated +/- treatment with a protein phosphatase, the experiments on the effects of stress have been repeated to generate total proteome data sets. These analyses have not substantially altered the conclusions of the paper, but have strengthened their support.

Reviewer #1:

Shepachev et al. describe TRAPP, a new method for the discovery of RNA-binding proteins (RBPs) via silica-based enrichment of crosslinked RNA-protein complexes. The method is validated by RNase/DNase treatment controls and applied to yeast and E. coli, including a stress condition in yeast. PAR-TRAPP is further introduced as a modified version that utilizes 4tU in order to refine the method, whereas iTRAPP allows the determination of RNA interaction sites on proteins. Finally, the authors pick yeast enolase 1, predicted to be an RBP by their assay, for downstream validation by CRAC, and show that this protein mostly binds cytoplasmic tRNAs. The method is described in sufficient technical detail and will be of great value for the field. Researchers could apply it even without MS analysis for a protein of interest, detecting that protein by, e.g., immunoblot. However, we recommend several issues to be addressed prior to publication.

We thank the referee for the careful evaluation of the work.

Abstract

How does the discovery of Enolase interaction with tRNAs fit to the sentence in the results part: 'Eno2, was previously reported to interact with mitochondrial tRNAs (Entelis, Brandina et al., 2006)'. If the discovery is limited to Eno1, this should be stated here.

The previous data addressed interactions for only Eno2, whereas we have data only for Eno1. However, Eno1 and Eno2 are quite closely related in sequence, so we mentioned this result in the text. In any event, we have removed these sentences from the revised Abstract and Results.

Introduction

p. 3, 1st paragraph: References would be helpful after the sentence: "Both approaches have been used extensively in many systems."

There are a very large number of suitable references, so we now cite a relevant review.

p. 3, 2nd paragraph: Double-check whether the references (Garcia-Moreno, Noerenberg et al., 2018, Gatto, Breckels et al., 2018, Trendel, Schwarzl et al., 2018) really fit with the extraction methods described.

We have corrected and updated the references.

p. 3, 3rd paragraph: The authors jump into listing the benefits of TRAPP techniques before describing what these are.

The text has been altered to very briefly outline the approach.

The authors claim that TRAPP can reveal the 'steady-state protein-RNA interactome'. This may be true for a relative comparison between conditions. Nevertheless, a precise determination of steady-state conditions requires a truly quantitative, unbiased crosslinking approach which is not the case here. The authors should revise this statement.

Text has been altered.

Results

p. 5, 2nd paragraph: The authors discuss the specificity for RNA by the UV crosslinking efficiency. In addition to this, an important step in the protocol is the ethanol-based wash in TRAPP which

washes away bound DNA (Avison M (2008) *Measuring Gene Expression*. London, UK: Taylor & Francis). The authors should add this point.

We thank the referee for pointing out this reference. We have included this and revised the text.

p. 5, 2nd paragraph: Could the authors please elaborate more on the quantity of 'some proteins'.

This comment refers to the gels shown in Figure S2C, in which a low level of residual staining is visible in the absence of UV crosslinking. We have altered the wording to make this clearer. We are not in a position to quantify the data with any accuracy. However, an indication of the suspected identity of these proteins is given by the subsequent iTRAPP data, and we have included mention of this on P5. Related to this, we also changed “some proteins” to “these proteins” on P13.

It would be helpful to see a GO-term based upon absolute quantification of RBPs, DNA-binding, and other proteins for validation of the method based on total lysate MS. In comparison to the total proteome, in TRAPP/PAR-TRAPP/iTRAPP the RBPs should be enriched. The enrichment could be taken into account by binning and the expected result would be an increased ratio of RBPs in bins with higher enrichment.

We agree that this would be useful but, unfortunately, we cannot readily generate these data. Protein abundance is absolute quantification, whereas SILAC gives enrichment over background. Since we do not have absolute quantification of proteins in TRAPP, we cannot state how the absolute abundance in the TRAPP proteome compares to the total proteome. Such data could be generated, by SILAC comparison of a lysate input to the TRAPP output, but the cost would be loss of the discrimination between crosslinked and non-crosslinked proteins.

In addition, the enrichment levels should be correlated to the total protein levels.

We have calculated Pearson correlation coefficients between fold enrichment and protein abundance for 1.4J TRAPP and PAR-TRAPP separately for UV enriched “RNA biology” GO term proteins and for all other UV enriched proteins. We did observe a very weak correlation in both TRAPP (Pearson 0.33) and PAR-TRAPP (Pearson 0.27) datasets for “RNA biology” GO terms proteins. This is presumably because rather abundant ribosomal proteins tend to score very high enrichment values with (PAR)-TRAPP, thus setting the trend. However, there was no correlation between fold enrichment and abundance amongst proteins not annotated with “RNA biology” GO terms in both PAR-TRAPP (Pearson 0.15) and PAR-TRAPP (Pearson -0.18). It is difficult to interpret these numbers. How well a given protein will be enriched depends on the signal strength (amount of protein in +UV sample) and on background noise floor (amount of protein in -UV sample). The former depends on the properties of RNA interaction for that protein, while the later depends on physical and chemical properties of the protein (i.e. interaction with silica, or crosslink-independent

ability to stick to RNA under purification conditions). We believe altogether these factors should not correlate with protein abundance, which is what we observe. We are not sure if this technical detail would be of interest to our readers.

p. 5, last paragraph: "Several reports ..." needs a reference.

Added

p. 6, last paragraph should be rephrased to clarity, it is hard to follow.

This section has been changed and relocated.

p. 7, 1st paragraph: The authors postulate that abundant proteins have acquired features that favor transient RNA interaction. This should be referenced or elaborated in detail. How is this different from the well-known unspecific crosslinking of RNA with abundant proteins at high concentration?

The SILAC quantitation allows us to confidently identify interactions that occur *in vivo*, and distinguish these from *in vitro* binding to RNA. The recovery of both non-specific and specific interactions will scale with protein abundance, but it was unclear why the number of copies of a protein would correlate with its propensity to crosslink with RNA. In any case, this model is given less prominence in the revised MS.

p.7: The authors should use percentage indications rather than absolute numbers, because the protein levels recovered between TRAPP and PAR-TRAPP are very different.

Giving the difference in recovery of "unexpected" RNA interactors TRAPP and PAR-TRAPP as a percentage of all hits is a bit misleading - as the difference in total numbers is predominately due to this type of protein. However, we have added the % of proteins to the text.

p. 7, 2nd paragraph: The authors should clarify the term 'dynamic range', and discuss the values below $\log_2 = 1$ compared to the pA data in Fig. 2B,C.

As both referees raised a question on the use of the term 'dynamic range', we have rewritten the section of text containing the term. Proteins below 2-fold enrichment are mostly not classified as RNA binding in PAR-TRAPP due to lack of statistical power at small enrichment values. Hence most of these proteins are labelled in black in the new figures 3B and 3C. It is difficult to compare our data to the polyA capture set, as it does not contain proteins that were detected, but were not enriched. So, we cannot comment on the performance of the two RBPome isolation methods for proteins with enrichment around 2 fold in +UVA. However, the TRAPP data are very reproducible and some statistically significant changes of less than 2 fold ($\log_2 < 1$) were observed.

p. 7, 3rd paragraph: The authors present a valuable data set for bacterial candidate RBPs but this area is neither properly introduced and referenced nor discussed in sufficient detail. Which RBPs would one expect to detect in *E. coli* or related bacteria, which of them were detected in the end? Are copy numbers known? This is the more important since the entire next paragraph is then devoted to proteins that have well-established primary functions unrelated to RNA-binding, so regardless of their detection here, one would rather consider them contaminants. For the proteins listed in the last sentence, enrichment factors and references are missing.

We have substantially altered this section of the text.

p. 8, 1st paragraph: The authors refer to a similar correlation between protein abundance and TRAPP enrichment in yeast and *E. coli*. Unfortunately, the bacterial data is not shown. A suppl. figure correlating protein abundance (MS of lysate input) and enrichment would be necessary for yeast and *E. coli* in order to draw valid conclusions here.

This has been included, together with an analysis of the effects of altered UV exposure (new Figure 4).

p. 10, 3rd paragraph and Fig. 4A: Only the differences in TRAPP enrichment are compared, but how does the protein level change upon stress? The authors should compare the lysate protein abundance between the stressed and non-stressed conditions to make sure that the enrichment they observe is not due to differing protein levels.

Data for the total proteome have been included in the revised MS (new Figure 5B).

p. 8, 1st paragraph: Fig4B should be 3B.

The figure numbers have now changed.

p. 10, 1st paragraph: The authors propose an effect of acid stress on early ribosomal biogenesis. How do the authors exclude an effect in translation that is propagated to ribosome biogenesis by a lack in the production of biogenesis factors? The authors should check for equal levels of biogenesis factors in both conditions to bolster their claim.

MS quantitation of the total proteome confirms that bulk changes in the abundance of ribosome synthesis factors does not occur.

p. 20: The authors should clarify how the SILAC based enrichment was determined. Were heavy and light samples been mixed prior to x-linking or was only the heavy sample x-linked? The text

does not clearly explain this, whereas in Fig1A it seems like the ¹³C-labeled cells were x-linked and the ¹²C-labeled cells were not.

Heavy and light cells were mixed after irradiation but prior to lysis. In all experiments label swaps were included to ensure that the isotopic labeling did not interfere with the results. We have altered the text to make this clearer (P4).

Other comments

p. 5, 2nd paragraph: The authors should rephrase this sentence: "Degradation of DNA had little effect on protein recovery, whereas this was substantially reduced by RNase treatment". It is not clear at which stage of TRAPP the protein recovery refers to.

This refers to the gel shown in the figure panel cited in the previous sentence. We have moved the callout to the figure.

p. 5, 3rd paragraph: The sentence about the 'missing' peptides should be rephrased.

The text has been altered

p. 12, 2nd paragraph: The second PO₄ should be indicated in the brackets.

Corrected

p. 13 and 14, Fig. 7 should be 6.

The figure numbers have now changed.

Fig. 1E: The top row should be a heading.

The top row is a heading. This has been by placing the header in bold text.

Fig. 1D: The green ring must be bigger to match the proportion of proteins. Add a circle for YMDB proteins that are compared in the text.

In Figure 1, the rings should be in proportion to the number of proteins, since they were drawn by a program with this setting. We have included a ring showing overlap between TRAPP, PAR-TRAPP and YMDB. It should, however, be noted that a number of tRNA biology proteins are found in YMDB, so this ring is not entirely clear of RNA biology proteins – although it predominately contains enzymes of general metabolism.

Fig. S2A: RNase and DNase should have a lower case 'a'.

Corrected

Fig. 3C,D: How are conserved RBPs and non-RBPs defined?

Figure 3C is now Figure 4F, Figure 3D has been omitted.

The legend to the figure states: "Pie chart of Inparanoid 8.0 database orthologous clusters between *S. cerevisiae* and *E. coli*. For a cluster to be labelled as conserved RNA interacting ("conserved, RBPs"), it was required to contain at least one bacterial and one yeast protein enriched in TRAPP. "Conserved, RBPs metabolic" are clusters where at least one protein in yeast or bacteria is identified in the YMDB or in ECMDB databases respectively (see materials and methods)."

Fig. 3D should be referred to in the main text. It would be important to also give GO-term processes for known RNA-binding proteins as a positive control to the newly identified ones.

This panel has been omitted from the revised text.

Reviewer #2:

In this manuscript, Shepachev et al introduce a generic method for the isolation and characterization of RNA-associated proteins. They achieved this by combining UV-crosslinking, silica-mediated isolation of RNA-protein conjugates, and mass spectrometry for protein identification. This method, termed TRAPP, is accompanied by a similar methodology where nascent RNA is labeled with 4-SU, followed by crosslinking at slightly higher wavelength, and further work-up as for TRAPP. The authors found TRAPP to be more sensitive (i.e. identifying more proteins), and they identify and partially confirm RNA-binding of novel proteins. Furthermore, they determined global changes in RNA-binding of acid-stressed cells, and identified RNA-crosslinking sites in many proteins in a procedure termed iTRAPP, which includes a TiO₂ enrichment step to isolate RNA-crosslinked peptides.

This manuscript follows a series of studies that have appeared over the last few years to characterize proteins that interact with poly-adenylated RNA, however now extending this to (non-coding) RNAs lacking a poly(A) tail. This is highly relevant given the often poorly characterized mechanisms that mediate functionality of RNA. However, the manuscript suffers from major shortcomings with regard to the validity and rigor of the used methodology, casting doubt on some of the chief findings of the paper.

We thank the referee for the time and effort expended on the review.

Major concerns:

1. The main concern about the TRAPP methodology, and primarily the used conditions for UV-irradiation, is that the data strongly indicate over-crosslinking. In particular, the observation that protein crosslinking scales with protein abundance (Figure 1g) is alarming: the authors should have recognized this as an emblematic indication of over-crosslinking, and thus of non-specific RNA-protein interactions, instead of postulating that 'abundant proteins have features that predispose them towards RNA association' (page 6), that 'highly abundant proteins have systematically acquired features that favor transient RNA interaction' (page 7), or, as a quite astonishing claim that there is no specificity after all in the way RNA interacts with protein, but that instead RNA and protein 'rather act biophysically "en masse"' (page 16). Indeed they are crosslinked 'en masse' - probably as an artifact.

We are unclear what the referee envisages as "over-crosslinking". This is not a term that we have previously encountered in the context of UV crosslinking and, as far as we can determine, it has not been used in the literature. Analyses using chemical crosslinkers, e.g. formaldehyde, are indeed potentially susceptible to over-crosslinking due to the tendency for interactions to be bridged. This is not, however, generally perceived to be an issue for UV crosslinking, which requires direct contact between the amino acid and nucleotide and generally has low efficiency.

As requested, we have repeated the TRAPP analyses, using a range of UV doses in both yeast and E.coli, and have included the data in the revised MS. From these analyses it is unclear what would be the "optimal" dose, since reducing the exposure lowers recovery of both annotated RNA-interacting proteins and other factors.

In fact the authors themselves provide additional evidence for over-crosslinking in TRAPP, by showing that the correlation between protein abundance and crosslinking to RNA does not exist in PAR-TRAPP, at the same time identifying half the number of proteins (Fig 1g): using 4-SU as the crosslinking base in PAR-TRAPP will exhaust crosslinking as soon as all 4-SU are photoactivated. In TRAPP, this seems not to be the case with 254 nm crosslinking where one simply starts crosslinking everything in the surrounding of RNA. In a way the authors propose this as a potential mechanism (page 15), without recognizing it as a pitfall.

We agree with referee that better results obtained with 4SU crosslinking in PAR-TRAPP. This led us to use this approach for iTRAPP and the stress analyses.

Over-crosslinking is likely the result of UV radiation at higher intensity than commonly used for RNA-interactome analysis, in the newly-constructed device (page 4-5, Fig S1E). Although

mathematically cells may have received the same cumulative dose in both cases, I could very well imagine that artifacts may be introduced when irradiating cells at high intensity. In fact, the absence of a correlation between RNA-interaction and protein abundance in previous RNA-interactome studies has been a strong argument for the specificity of the observed interactions.

Unfortunately, this comment appears to be entirely based on a misreading of the MS. The newly constructed device was used for UVA irradiation in PAR-TRAPP, not for TRAPP. This was clearly explained in the results, figure legend and methods.

Collectively, as a result of this, it is highly questionable how the results obtained by TRAPP (Fig 1, 3 and all associated supplemental material) can be trusted. To remedy this fundamental issue, the authors should titrate the UV radiation dose, monitor when high abundance proteins start to disappear, and then accordingly adjust the radiation intensity and duration to minimize background caused by non-specific interactions.

As noted above, we have included a range of UVC doses in the revised MS.

2. The iTRAPP workflow includes a TiO₂ enrichment step to purify RNA-crosslinked peptides (Fig 5). Surprisingly, only 48% of the observed crosslinking sites fall within known RNA-binding domains, which contrasts with previous studies who find >85% (e.g. Kramer et al).

Inspection of published structures for RNP complexes (e.g. ribosomes, pre-ribosomes, spliceosomes, snRNPs, snoRNPs) reveal that close RNA-protein contact are found with known RNA-binding domains and with many other regions. This result would therefore have been strongly predicted.

In addition, the authors find that RNA-protein crosslinks strongly correlate with sites of phosphorylation (Figure 5e). Both observations may be explained by the likelihood that in fact TiO₂ strongly enriched for phospho-peptides (which can be easily tested by including phosphorylation as a PTM in the database search). This would not be surprising at all if the authors omitted a phosphatase step, meaning that inclusion of such a treatment should result in a distinct improvement of the results.

As requested, we have repeated the iTRAPP analysis including lambda phosphatase treatment, which has preference for phospho-peptides over nucleotides as substrates. Following this treatment phospho-protein recovery was reduced 10-fold, but nucleotide di-phosphates remained the predominant modification.

Moreover, the referee's proposal requires that the mapping software systematically mis-identified peptides carrying a nucleotide mono-phosphate plus a second phosphorylation as carrying a nucleotide di-phosphate. As mentioned in the revised Discussion, this is highly improbable. More

generally, we would strongly expect protein phosphorylation to be anti-correlated with neighboring RNA crosslinking, due to charge repulsion.

If necessary, we can include the list of phospho-peptides identified from the RNA-bound samples. There is, however, a caveat with these data, in that the iTRAPP analyses were not performed with SILAC quantitation. It therefore remains possible that some of the phospho-peptides recovered were derived from proteins retained on the silica beads that were not crosslinked in vivo.

Other concerns:

3. TRAPP critically relies on silica enrichment of RNA-crosslinked proteins. Yet, proteins or RNA-crosslinked proteins are usually not retained by silica - do the authors have a proposed mechanism how this may still occur?

We are unsure of the basis for this statement by the referee. The use of silica for RNP purification has recently been independently reported (Asencio, Chatterjee and Hetze, Life Sci. Alliance, 2018). More generally, it has long been known that nucleic acids can be retained on silica while proteins remain unbound. Indeed, this is the basis of commercial miniprep kits. Notably, however, in our hands miniprep columns do not give good results in TRAPP, possibly due to the formulation of silica used.

In particular, how would a (large) protein crosslinked to a (small) RNA be retained, while the same free protein would not?

We are unclear what conceptual difficulty the referee envisages. The RNA acts as an affinity tag retaining the protein on the silica matrix. This principle seems well established; e.g. a His6 tag allows retention of proteins on Ni²⁺ columns to which they do not otherwise bind efficiently.

4. Fig 2b and c raise a couple of questions: i) the figure is hard to understand without declaring Si and pA.

Si has been replaced with PT, which is defined. We feel that pA will be evident to the reader.

ii) the difference between panels b and c is unclear.

The panels were clearly labeled in the figure. Panel B shows a comparison of only the subset of proteins that were in common between PAR-TRAPP and polyA capture; panel C compares all proteins recovered with each technique. We have altered the legend to make this clearer

iii) Authors claim that GO terms like RNA binding, translation etc have a higher dynamic range (page 7). This is not true, instead they have a larger spread in their enrichment.

As both referees raised a question on the use of the term ‘dynamic range’, we decided to rewrite the part of text containing the term.

iv) It is unclear how black dots in TRAPP (i.e. not passing significance) are still classified as TRAPP-enriched. Similarly, some colored dots are apparently significant, although showing a fold change <1 , or some even <0 (i.e. depleted?).

As noted above, the TRAPP data are very reproducible and statistically significant of less than 2 fold ($\log_2 <1$) were indeed observed. As the referee notes, some proteins were reproducibly depleted following UV irradiation. We attribute this to crosslinking to insoluble components, e.g. membranes, and mention this in the revised text.

Changes not passing statistical significance were not included in data analysis, although we do point out specific, seemingly relevant, examples in the ribosome synthesis pathway, with appropriate caveats.

v) in general, the scatter plots as shown in fig 2bc and elsewhere in the paper are only partially informative since they only show ratios, not significance. Therefore they should be replaced by volcano plots to indicate both parameters.

Unfortunately, it is not feasible to present these comparisons in the figure, as 32 volcano plots for each of the GO term category. We therefore added volcano plots representing all quantified proteins for each of the TRAPP and PAR-TRAPP experiment (new Figure S4).

5. Figure 3a: why are so many proteins identified in the negative control (-UV), or even enriched there (e.g. membrane proteins)?

As discussed in the text, many proteins retain some RNA binding affinity even when denatured. For this reason, we feel that the inclusion of SILAC labelling, although not generally used in the field, is very important for the quantitative analysis.

6. In their analysis of dynamic changes in RNA-protein interactions (page 8) the authors prefer to use PAR-TRAPP over TRAPP. The reasoning is a bit surprising after their prior claims that TRAPP outperforms PAR-TRAPP. However, the lack of dynamic range in the observed stress-induced interaction differences in TRAPP is fully understandable in the light of over-crosslinking: all RNA-protein crosslinks are saturated, both in the absence and presence of stress, hence levelling out any differences that may exist.

This appears to be based on a misreading of the text. We are unclear where the referee feels that we claimed that TRAPP out-performs PAR-TRAPP. This was not the conclusion that we drew from the initial analyses and we have altered the text to make this more explicit.

There may also have been some misunderstanding of the efficiency of UV induced crosslinking. As far as we are aware crosslinking efficiencies, even for stably bound RNA-protein complexes, have never been reported to exceed single digit percentages. We are therefore unclear what the referee intends by the suggestion that crosslinks are saturated, which might appear to imply close to 100% crosslinking.

7. It is unclear why the authors propose a defect in pre-60S ribosomes (page 10), and not a functional change?

Ribosome synthesis is energetically expensive and blocked by many different stresses. We propose that the altered RNA-protein interactions reflect functional changes that block 60S maturation. We have replaced “defect” with “block” (P10).

8. For iTRAPP, 10 liters of yeast culture was used. Since this is a very large amount, it will be useful if the authors can comment why they need this large volume (especially compared to other methods), or at which steps (Fig 5a) the major losses are experienced.

We initially used a pre-made preparation of frozen, crosslinked cells that happened to be from a 10L culture. Subsequent experiments used 2.5L cultures, but we have not attempted to determine the minimum amount needed for iTRAPP analysis. We would assume that substantially less culture could be used if necessary.

Two other methods have been reported for the identification of crosslinking aminoacids, by ourselves (Peil et al., 2018) and Kramer et al. (2014). Both analyses combined the results of multiple different experiments to generate the final lists of interactions, and therefore used substantially more material than iTRAPP.

9. The discussion contains a long section on RNA-mediated phase separation. Although this is a popular topic, the section is highly speculative, and none of what is discussed is based on data produced or shown in the paper. It should thus be eliminated.

We have attempted to make this section more compact, but would contend that our results should be discussed and placed in context. This is our best interpretation and we feel that it will be of use to the readership.

10. Page 12: it is unclear how the authors derive crosslinking efficiency for each amino acid.

Specifically, in many peptides (Table S8) most if not all amino acids score very low (<0.1) for localisation of crosslinks, i.e. I assume that this means that no crosslink site could be determined. First question is why this is, second how this is handled in data interpretation, i.e. on which basis were peptides in/excluded for scoring as in Fig 5C. For instance, if a W in a 15-amino acid peptide scores 0.2 for its crosslinking likelihood (table S8), is it considered crosslinked or not? And how is the value (0.2) propagated to calculate a 'fold change' (Fig 5c)?

In the revised text we consider only sites at which the spectra provided clear evidence for a crosslink at a single position.

The link-site score is a likelihood score, reported in table S8. The score reflects how much of the ms2 intensity cannot be explained by a link-site that would be explainable by another link-site. Such unexplained MS2 intensity is nearly always present. Since Xi allows neutral water and ammonia losses, the longer the peptide, the higher the chance that some of the unexplained MS2 intensity could be explained by a crosslink at another position within the peptide. Therefore, the longer the peptide, the lower the individual amino acid crosslink scores tend to be. In our analysis the amino acid having the highest score is considered crosslinked, regardless of how low/high the score is. We have reprocessed the data in order to make sure the ambiguous cases, in which 2 or more amino acids would share the highest crosslink score, are not included into the new Figure 6C. Although we cannot be certain that every crosslink mapped in our analysis is correct, we believe that collectively they are a good representation of amino acid crosslink frequencies.

11. On page 17 the authors say: 'Only around 10% of RNA-interacting proteins showed differences in crosslinking of greater than 2-fold, suggesting that the stress responses are relatively specific'. I cannot follow the logic of this - instead I would interpret the observation that the response is mild, not specific.

The possibility existed that cellular stress due to altered intercellular pH might result in very widespread changes in RNA-protein interaction. It has been reported that these conditions result in "solidification" of the cytoplasm (Munder et al., eLife, 2016). Indeed, it was this observation that initially prompted our use of sorbic acid. However, this was not what we found, with most proteins unchanged and a small number showing increased or decreased binding.

12. On page 18 the authors say: 'We speculate that stress signaling allows abundant proteins of intermediary metabolism to form more stable interactions with RNA, allowing PAR-TRAPP detection'. It is unclear what this speculation is based on, and actually what it means: How can protein abundance, as a biophysical property, induce formation of 'more stable' interactions?

We now include reference to the paper by Munder et al. (2016). We had omitted this discussion on grounds of space.

13. In several instances the authors draw far-reaching conclusions where more simple explanations seem more obvious.

For instance: i) Page 13: Eno1 was identified to interact with mRNA encoding Eno2. Since these are among the most abundant protein and RNA, respectively, in a cell, this strongly suggests a coincidental interaction.

We are unclear what point the referee is making here. We mentioned the interaction of Eno1 with the *ENO2* mRNA in the text and then went on to explain why we do not think that this is functionally important. The suggestion that the *ENO2* RNA is among the most abundant in the cell seems odd. The RNA is at least 10,000-fold less abundant than rRNA. However, we have removed mention of the *ENO2* mRNA from the revised text.

ii) In addition, the authors find that Eno1 interacts with cytoplasmic and not mitochondrial tRNAs, leading them to conclude that 'these reflect bona fide, in vivo interactions' (page 14). An easier and more likely interpretation is that, as is known, Eno1 is localized in the cytoplasm and not in mitochondria.

We mentioned this point because, as described in the text, the closely related protein Eno2, which is also cytoplasmic, has been reported to bind mitochondrial tRNAs (Entelis et al. Genes Dev., 2006).

iii) Eno1 was found to interact with the T-loop, anti-codon or extended variable loop in tRNA (page 14), leading the authors to conclude that 'Eno1 binds to specific structural elements'. Instead, it is more likely that these are the only sites in a tRNA that can crosslink because the other ones are engaged in base-pairing in the tRNA itself. This may be verified in a CLIP experiment (under conventional crosslinking conditions) for an amino-acyl-tRNA synthetase.

We are unclear of the basis for this statement. The major crosslinks were found with the T-loop, whereas almost none were found with the D-loop. The most exposed loop in the tRNA is, of course, the anticodon loop. We recovered interactions here, but at much lower frequency than for the T-loop.

We assume the comment about "under conventional crosslinking conditions" is related to the misunderstanding noted above, concerning the crosslinking equipment. These conditions have been used in >50 publications.

To conclude this, the authors say that 'There is every reason to think that these data capture genuine, in vivo RNA-protein contacts'. Apart from the question if stylistically this is a fitting sentence, I wonder if it is actually true.

We are unclear what aspect of the sentence the referee is questioning. In contrast to analyses in which proteins recovered after crosslinking are compared to input, which the referee may have encountered, the SILAC quantitation allows us to confidently identify interactions that occur *in vivo*, and distinguish these from *in vitro* binding to RNA.

2nd Editorial Decision

25th February 2019

Thank you again for submitting your work to Molecular Systems Biology. We have now heard back from reviewer #2 who was asked to evaluate your study. As you will see below, reviewer #2 thinks that the study has improved as a result of the performed revisions. S/he lists however some remaining concerns, which we would like to offer you a chance to address in a revision.

REFEREE REPORTS

Reviewer #2:

In this revised manuscript, the authors have performed a number of experiments that have resolved some issues. However other comments were ignored and therefore the same concerns remain. My comments below follow the numbering in my initial report:

Point 1

> By over-crosslinking I meant crosslinking between other molecular species than protein-RNA. Most prominently this may include protein-DNA interactions. Although UV crosslinking efficiency is lower for DNA, this cannot be excluded. I appreciate that the authors have taken measures to avoid contamination introduced by DNA-protein crosslinks, the authors should demonstrate (instead of infer, Figure EV2D) that contamination by DNA is completely eliminated.

> An unexplained and disturbing observation remains that in TRAPP protein crosslinking scales with protein abundance, now shown to occur also at low UV dose (Fig 2D). This is and remains one of my main concerns, which was entirely disregarded in the authors' response. I'll therefore reiterate here that explanations like 'abundant proteins have features that predispose them towards RNA association' and 'many of the novel RNA-protein interactions found in TRAPP do not have specific, pair-wise functional roles, but rather act biophysically "en masse" to modulate cell organization' are highly speculative and in contradiction with existing literature. Moreover, what 'features' do the authors refer to, and how can the 'en masse' model (negating specificity in RNA-protein binding) be married with the central view in the field (and a premise in this paper) that RNA-protein interactions are specific? This begs for a better explanation to exclude experimental bias before concluding on biology. And in case of the latter, this should be independently validated. In response to Reviewer 1 the authors state that this 'model' received less prominence, however a change of wording may not be sufficient to eliminate a potentially fundamental underlying bias.

> The authors agreed that better results were obtained with 4SU crosslinking in PAR-TRAPP, therefore I wonder what value and application areas they see for TRAPP?

Point 4, iv

> I do not see where/how the text was changed. Even then I do not understand how the authors attribute this phenomenon can this be explained by crosslinking to membranes, if the authors claim that their method is specific for RNA-protein interactions? Both claims cannot be true at the same time.

Point 12

> The reference has not been included in the main text. Moreover, the topic of the Munder paper is a different one and thus does not answer my point. Therefore the question remains: How can protein abundance, as a biophysical property, induce formation of 'more stable' interactions? This also directly relates to my comments under point 1.

Reviewer #2:

In this revised manuscript, the authors have performed a number of experiments that have resolved some issues. However other comments were ignored and therefore the same concerns remain. My comments below follow the numbering in my initial report:

1: Point 1

> By over-crosslinking I meant crosslinking between other molecular species than protein-RNA. Most prominently this may include protein-DNA interactions. Although UV crosslinking efficiency is lower for DNA, this cannot be excluded. I appreciate that the authors have taken measures to avoid contamination introduced by DNA-protein crosslinks, the authors should demonstrate (instead of infer, Figure EV2D) that contamination by DNA is completely eliminated.

This comment appears to be based on a misreading of the MS. Based on the data in Figure EV2D we do not infer that no DNA is recovered in TRAPP and make no claim that this is the case elsewhere in the MS. We wrote: “We conclude that the TRAPP protocol predominately recovers RNA-bound proteins” and this is the case. Demonstrating the complete elimination of any component would always be challenging and we cannot do it in this case.

2.1: > An unexplained and disturbing observation remains that in TRAPP protein crosslinking scales with protein abundance, now shown to occur also at low UV dose (Fig 2D). This is and remains one of my main concerns, which was entirely disregarded in the authors' response.

We agree with the reviewer that this is unexpected. We were, however, disappointed that the referee feels we did not address this issue in the revised MS. We expended a great deal of time and resources in additional work to address the relationship between UV dose and protein recovery. These analyses doubled the amount of primary data present in the revised MS relative to original version.

As the reviewer points out, we cannot exclude the possibility that this result arises from some underlying bias in the method, but nor can we discount the possibility that these proteins are genuine RNA-binders. Thus, the current version of the manuscript describes both possibilities.

2.2: I'll therefore reiterate here that explanations like 'abundant proteins have features that predispose them towards RNA association' and 'many of the novel RNA-protein interactions found in TRAPP do not have specific, pair-wise functional roles, but rather act biophysically "en masse" to modulate cell organization' are highly speculative and in contradiction with existing literature.

These statements have been removed from the Results and Discussion, respectively, in the revised MS.

2.3: Moreover, what 'features' do the authors refer to, and how can the 'en masse' model (negating specificity in RNA-protein binding) be married with the central view in the field (and a premise in this paper) that RNA-protein interactions are specific?

We would contest the opinion that all RNA:protein interactions are necessarily specific. Several groups have independently reported that multiple proteins not predicted to have any direct role in RNA metabolism (e.g. glycolytic enzymes) are recovered following UV crosslinking; whether by poly(A) selection, phase separation (eg OOPS) or TRAPP. Indeed, while this manuscript was under revision, very recent publications have confirmed this general finding (Queiroz, Smith et al., 2019, Trendel, Schwarzl et al., 2019). While many RNA-protein interactions undoubtedly are site-specific, this seems very unlikely to be the case for all proteins recovered in our, or other, analyses. Indeed, the term “enigmRBPs” has been coined to cover the many RNA-interacting proteins that lack any known function in RNA biology (Beckmann, Horos et al., 2015). We include this reference in the revised text (P8). In the case of the glycolytic and other metabolic enzymes, an explicit model has been proposed for phase separation driven by RNA interactions (Castello, Hentze et al., 2015) and we now reference this (P18).

2.4: This begs for a better explanation to exclude experimental bias before concluding on biology. And in case of the latter, this should be independently validated. In response to Reviewer 1 the authors state that this 'model' received less prominence, however a change of wording may not be sufficient to eliminate a potentially fundamental underlying bias.

As noted above, we have further modified the Results and Discussion.

3: > The authors agreed that better results were obtained with 4SU crosslinking in PAR-TRAPP, therefore I wonder what value and application areas they see for TRAPP?

While PAR-TRAPP seems to yield better results, it will not always be feasible. For example, as reported in this manuscript, *E. coli* is unsuited for 4SU labeling we therefore performed only TRAPP in this species. TRAPP will also be needed in many other systems for which 4SU labeling is not feasible, including plants and animal tissues.

4: Point 4, iv

> I do not see where/how the text was changed. Even then I do not understand how the authors attribute this phenomenon can this be explained by crosslinking to membranes, if the authors claim that their method is specific for RNA-protein interactions? Both claims cannot be true at the same time.

UV irradiation can induce protein crosslinking to other macromolecules, including lipids, but these should not be recovered by TRAPP as silica enrichment is specific for nucleic acids.

5.1: Point 12

> The reference has not been included in the main text. Moreover, the topic of the Munder paper is a different one and thus does not answer my point.

The reference was cited and discussed on P18 of the MS. We have highlighted this more in the revised text (PP 19 and 20).

5.2: Therefore the question remains: How can protein abundance, as a biophysical property, induce formation of 'more stable' interactions? This also directly relates to my comments under point 1.

We do not make this statement in the MS. However, as stated above, we have now altered the Results and Discussion regarding protein abundance and RNA binding.

References:

Beckmann BM, Horos R, Fischer B, Castello A, Eichelbaum K, Alleaume A-M, Schwarzl T, Curk T, Foehr S, Huber W, Krijgsveld J, Hentze MW (2015) The RNA-binding proteomes from yeast to man harbour conserved enigmRBPs. *Nature Comm* 6: 10127

Castello A, Hentze MW, Preiss T (2015) Metabolic Enzymes Enjoying New Partnerships as RNA-Binding Proteins. *Trends Endocrinol Metab* 26: 746-757

Queiroz RML, Smith T, Villanueva E, Marti-Solano M, Monti M, Pizzinga M, Mirea D-M, Ramakrishna M, Harvey RF, Dezi V, Thomas GH, Willis AE, Lilley KS (2019) Comprehensive identification of RNA–protein interactions in any organism using orthogonal organic phase separation (OOPS). *Nature Biotech*

Trendel J, Schwarzl T, Horos R, Prakash A, Bateman A, Hentze MW, Krijgsveld J (2019) The Human RNA-Binding Proteome and Its Dynamics during Translational Arrest. *Cell* 176: 391-403.e19

Accepted

13th March 2019

Thank you again for sending us your revised manuscript. We are now satisfied with the modifications made and I am pleased to inform you that your paper has been accepted for publication.

Corresponding Author Name: Prof. David Tollervey -

Manuscript Number: MSB-18-8689RR